# Diurnal variation of high-level clouds from the synergy of AIRS and IASI space-borne infrared sounders

Artem G. Feofilov and Claudia J. Stubenrauch

LMD/IPSL, Sorbonne Université, UPMC Univ Paris 06, CNRS, École polytechnique, Palaiseau, 91128, France

*Correspondence to*: A. G. Feofilov (artem. feofilov@lmd. polytechnique. fr)

**Abstract.** By covering about 30% of the Earth and by exerting a strong greenhouse effect, high-level clouds play an important role in the energy balance of our planet. Their warming and cooling effects within the atmosphere strongly depend on their emissivity. The combination of cloud data from two space-borne infrared sounders, the Atmospheric InfraRed Sounder, AIRS, and the Infrared Atmospheric Sounding Interferometer, IASI, which observe the Earth at four local times per day, allows us

to investigate the diurnal variation of these high-level clouds by distinguishing between high opaque, cirrus, and thin cirrus clouds. We demonstrate that the diurnal phase and amplitude of high-level clouds can be estimated from these measurements with an uncertainty of 1.5 h and 20%, respectively. By applying the developed methodology to AIRS and IASI cloud observations for the period of 2008−2015, we obtained monthly geographical distributions of diurnal phase and amplitude at a spatial resolution of 1° latitude x 1° longitude. In agreement with other studies, the diurnal cycle of high-level clouds is the

largest over land in the tropics. At higher latitudes, their diurnal cycle is the largest during the summer. For selected continental regions we found diurnal amplitudes of cloud amount of about 7 % for high opaque clouds and for thin cirrus, and 9 % for cirrus. Over ocean, these values are 2 to 3 times smaller. The diurnal cycle of tropical thin cirrus seems to be similar over land and over ocean, with a minimum in the morning (9h LT) and a maximum during night (1h LT). Tropical high opaque clouds have a maximum in the evening (21h LT over land), a few hours after the peak of convective rain. This lag can be explained

by the fact that this cloud type not only includes the convective cores, but also part of the thicker anvils. Tropical cirrus show a maximum amount during night (1h LT over land). This lag indicates that they are part of the deep convective cloud systems. However, the peak local times also vary regionally. We are providing a global monthly database of detected diurnal cycle amplitude and phase for each of these three high-level cloud types.

## 1    Introduction

Due to the importance of clouds for the Earth's energy budget, global satellite observations of cloud properties and their diurnal variations are essential for climate studies, for constraining climate models, and for evaluating cloud parameterizations. The diurnal variation of clouds modulates the radiative cooling and heating of the atmosphere and of the surface. Both the clouds embedded in the planetary boundary layer and the clouds connected with the surface through deep convection exhibit systematic diurnal variations related to the daily cycle of surface solar heating.

The International Satellite Cloud Climatology Project (ISCCP, Rossow and Schiffer, 1999) uses multi-spectral imager data from a combination of polar orbiting and geostationary weather satellites to for a globally complete long-term cloud data record at spatial and temporal scales consistent with cloud dynamical processes (approximately 3 hr, 25 km). Although many regional studies were done earlier, the first global analyses of diurnal cloud variations were based on the ISCCP products (Cairns 1995, Rossow and Cairns, 1995). Based on these results, the most notable features of the cloud diurnal cycle are significant differences between the phase of diurnal low-level cloud variations over ocean and land and between the phase of diurnal low-level and high-level cloud variations:

- low-level clouds over ocean have a maximum amount early morning, while over land the maximum is in the early afternoon;

- high-level clouds have a maximum amount early to late evening;

- mid-level clouds have a maximum amount late at night or early morning.

However, the combination of maximal two atmospheric window channels, one IR and one visible, (the latter not working during night) leads to a low sensitivity of ISCCP to thin cirrus at night and when low-level clouds are underneath.

By contrast, the high spectral resolution of the IR vertical sounders used in this study allows to select the spectral channels with the contribution functions centred at different heights: the radiances measured near the centre of the $15\mu m$ $CO_2$ absorption band are sensitive to the upper atmospheric layers while the radiances in the absorption band wings are used to probe successively lower levels. Compared to other passive remote sensing instruments, IR sounders are sensitive to cirrus with emissivity as low as 0.1, day and night (Stubenrauch et al. 2010, 2017; Menzel et al., 2016). TIROS-N Operational Vertical Sounder (TOVS) data (TOVS Path-B, Scott et al. 1999) have been used by Stubenrauch et al. (2006) to identify high opaque clouds, cirrus, and thin cirrus according to their emissivity, and by exploiting the time drifting of the afternoon polar orbiting NOAA satellites, in combination with the non-drifting morning orbits, concluded over land in the tropics and midlatitude summertime:

- high opaque clouds have a maximum amount in the evening;

- thin cirrus increase during the afternoon and persist during the night;

- the varying proportions of thinner and thicker cirrus imply a gradual thickening of the cirrus clouds from late afternoon into the night time;

- mid-level cloud amount exhibits a small increase during night time.

As passive instruments are only able to provide information on the uppermost cloud layer in the case of multi-level cloud fields, the results on the lower cloud diurnal cycle will be inevitably modulated by the clouds above. Therefore, we concentrate on the diurnal variation of high-level clouds.

The CIRS (Clouds from Infrared Sounders) cloud climatologies established from the Atmospheric InfraRed Sounder (AIRS, Chahine et al., 2006) and the Infrared Atmospheric Sounding Interferometer (IASI, Hilton et al., 2012), now covering 15 and 10 years, respectively, have been presented by Stubenrauch et al.(2017). In this article, we use the synergy of these two instruments, observing each point of the Earth at least at four local times, to build a data base of amplitude and phase of the diurnal cycle of amount and emissivity of high-level clouds, which can further be used for regional and global climate studies and for climate model evaluation.

The structure of the article is as follows. In Section 2, we shortly describe the AIRS and IASI cloud data as well as the environmental data used for this study. Then we present the newly developed approach to estimate the diurnal cycle of cloud amount from a combination of AIRS and IASI observations. Section 3 first presents a comparison of the diurnal variation of high-level cloud amount from our method with results from other datasets (3.1). Then we introduce the diurnal variation separately of high opaque cloud, cirrus, and thin cirrus amount. For specific land regions, we try to establish the links and temporal lags between the different high-level cloud types, the surface temperature, and relative humidity (Section 3.3). Section 3.4 presents another application: as the combined dataset covers a period of eight years we analyse the geographical patterns of amplitude change as function of the global surface temperature change. Conclusions are drawn in Section 4.

## 2    Datasets and Methods

### 2.1    Cloud properties from AIRS and IASI

Since 2002, the AIRS cross-track scanning instrument aboard the polar orbiting Aqua satellite has been providing very high spectral resolution measurements of atmospheric radiation in 2378 spectral bands in the thermal infrared (3.74−15.40 μm), at a spatial resolution of about 13.5 km × 21 km at nadir to 41 km × 21 km at the scan extremes (Chahine et al., 2006). Local observation times (LT) are 1:30 and 13:30 LT.

IASI aboard the polar orbiting Metop-A platform is a Fourier Transform Spectrometer based on a Michelson interferometer, which covers the IR spectral domain from 3.62 to 15.5 $\mu$m. As a cross-track scanner, the swath corresponds to 30 ground fields per scan, each of these measures a 2 × 2 array of footprints. The geometry of IASI observations is similar to that of the AIRS instrument: ±48.3° ground coverage, 12 km resolution at nadir, with observations at 9:30 and 21:30 LT, since 2007.

The CIRS cloud property retrieval package (Feofilov and Stubenrauch, 2017; Stubenrauch et al., 2017) is based on a weighted $\chi^2$ method using eight channels along the 15 μm $CO_2$ absorption band (Stubenrauch et al., 1999b). It provides cloud pressure ($p_{cld}$), cloud emissivity ($\varepsilon_{cld}$), cloud temperature ($T_{cld}$) and cloud height ($z_{cld}$), as well as their uncertainties. We define the cloud types according to $p_{cld}$ and $\varepsilon_{cld}$: high-level clouds are the ones with $p_{cld} < 440$ hPa, and these are further divided into high opaque ($\varepsilon_{cld} > 0.95$), cirrus ($0.95 > \varepsilon_{cld} > 0.5$), and thin cirrus ($0.5 > \varepsilon_{cld} > 0.1$). Ancillary data (surface temperature, atmospheric temperature and water vapour) are used in the radiative transfer calculations of the retrieval. While the sensitivity of the retrieved cloud properties to ancillary data is small for high-level clouds, the low-level cloud amount is sensitive to surface temperature used in the retrieval (Stubenrauch et al., 2017). To avoid potential retrieval problems associated with inconsistent

ancillary data between AIRS and IASI, we have adopted the same ancillary dataset for both, namely, the ERA-Interim meteorological reanalysis (Dee et al., 2011) by the European Centre for Medium-Range Weather Forecasts (ECMWF), given 6-hourly at universal time. For the cloud retrieval, the ERA-Interim surface temperature and pressure as well as atmospheric temperature and water vapour profiles have been interpolated towards the local observation times of AIRS and IASI.

From Fig. 1, presenting latitudinal distributions of high opaque, cirrus and thin cirrus amount for January and for July, averaged from 2008 to 2015, separately at 01:30, 09:30, 13:30, and 21:30 LT, we deduce that (i) in the tropics (30S−30N) high-level clouds are present more than in half of the observations (the sum of the three cloud types reaches 60%), (ii) tropical high-level cloud amount maximum moves with season towards the summer hemisphere, while the amount of high-level clouds in the midlatitudes is larger in winter, due to storm tracks, and (iii) all cloud types demonstrate a diurnal variation, but its zonal

amplitude is small compared to the zonal mean of the corresponding cloud amount. It is the largest around the peak of the InterTropical Convergence Zone (ITCZ), with about 5% for cirrus. Fig. 1 will serve as a reference when considering the diurnal amplitudes discussed below.

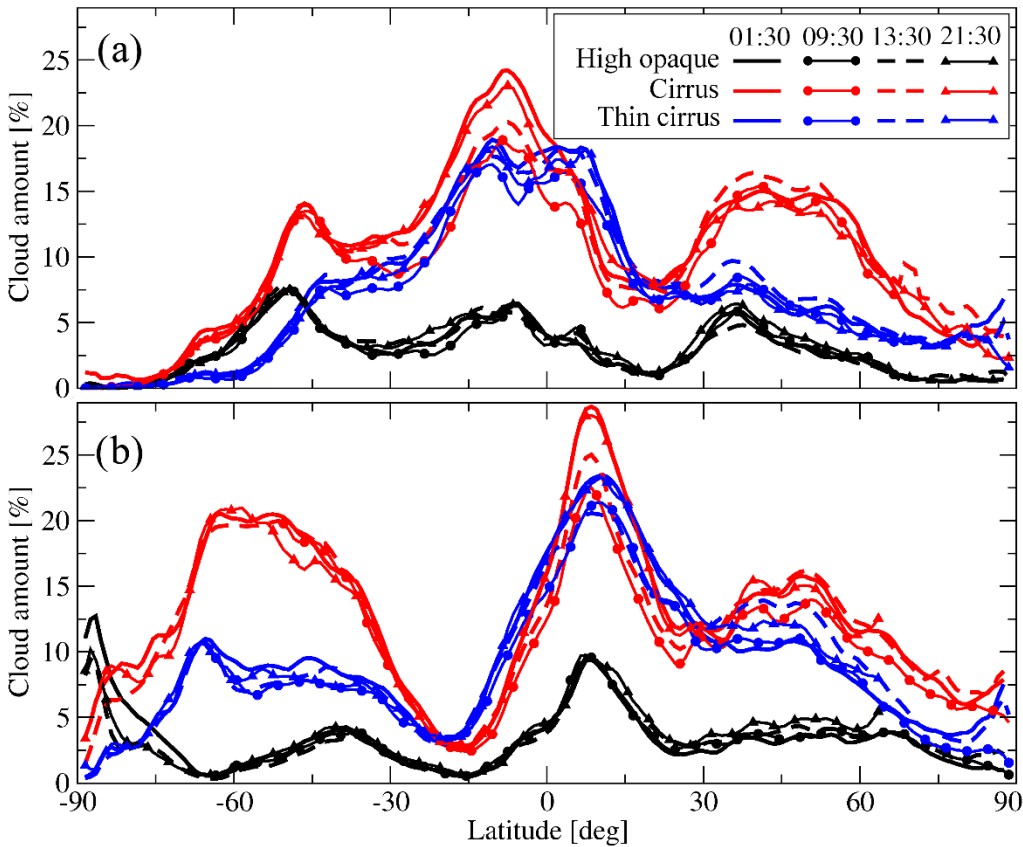

**Fig. 1. Latitudinal distribution of high opaque, cirrus, and thin cirrus cloud amount estimated from AIRS (01:30 and 13:30 LT) and**
**IASI (09:30 and 21:30 LT) by the CIRS retrieval: (a) January; (b) July. Climatological averages over 2008 to 2015.**

## 2.2 Estimating the diurnal cycle amplitude and phase

In this section, we develop an approach to identify both the amplitude and the phase (or the "peak time") of the diurnal variation of cloud amount, using a combination of AIRS and IASI cloud data, with four measurements per day. Both amplitude and phase depend on the cloud type, region, and season (Cairns, 1995; Soden, 2000; Tian et al., 2004; Stubenrauch et al., 2006; Eastman and Warren, 2014, and references therein), so for each location they should be determined individually.

The Nyquist-Shannon-Kotelnikov sampling theorem says: "if a function x(t) contains no frequencies higher than B hertz, it is completely determined by giving its ordinates at a series of points spaced 1/(2B) seconds apart". In application to diurnal variation analysis this means that four measurements per day are just on the edge of the diurnal cycle detectability. In addition, the condition of the theorem is not completely fulfilled since variations in cloud amounts are known to include variations on both diurnal and semi-diurnal time scales (e. g. Cairns, 1995), which is clearly beyond the detection limit. Moreover, the sampling of AIRS and IASI measurements is not equidistant in time with its 8 and 4 hour intervals. Correspondingly, one has to use an external source of information to ensure an unambiguous detection of the diurnal cycle and estimate its phase φ and amplitude $A$. We found this missing piece of the puzzle in the function describing the general behaviour of the diurnal cloud amount variation as a mixture of two harmonics, diurnal and semi-diurnal, as demonstrated by the analysis of ISCCP observations (Cairns, 1995). Accordingly the diurnal cycles of high-, middle-, and low-level clouds are well represented by a mixture of two harmonic functions of the following form:

$$A(t) = A_{24} \cdot \sin\left(\frac{2\pi}{24}t + \varphi_{24}\right) + A_{12} \cdot \sin\left(\frac{2\pi}{12}t + \varphi_{24} + \Delta\varphi\right) = A_{24} \cdot \left[\sin\left(\frac{2\pi}{24}t + \varphi_{24}\right) + 0.28 \cdot \sin\left(\frac{2\pi}{12}t + \varphi_{24} + \Delta\varphi\right)\right] \quad (1)$$

where the indices "24" and "12" correspond to diurnal and semi-diurnal harmonics, respectively, $t$ is time in hours, $\Delta\varphi$ is the phase shift between semi-diurnal and diurnal harmonics, and the numeric parameters are estimated from Fig. 1 of (Cairns 1995). It is interesting to note that a similar mixture of diurnal and semi-diurnal harmonics describes tropical precipitation (Bowman et al., 2005). Since the ratio of $A_{12}/A_{24} \approx 0.28$ obtained from (Cairns 1995) does not change much with the type of the cloud, we simplify the equation to a form shown in the second part of Eq. (1) and use this ratio throughout the analysis assuming that it is $A_{24}$, which dominates the diurnal variation. By analysing Fig. 1 of (Cairns, 1995) using least-square fitting of Eq. 1 we found that the phase shift $\Delta\varphi$ is equal to −2 h for the low- and mid-level clouds and to 0 h for high-level clouds.

With the $A_{12}/A_{24}$ set to 0.28 and $\Delta\varphi$ set to 0, the diurnal "shape" of Eq. (1) is fixed (see the grey line in Fig. 2a) and the problem reduces to one of determining the amplitude $A_{24}$ and $\varphi_{24}$. Two satellite instruments provide us with measurements four times of the day, and we determine the best fit amplitude and phase using a minimization technique based on the "sliding profile" approach as depicted in Fig. 2 and described below. This approach is similar to the one used in (Goldberg et al., 2013) where it was applied to determine the phase and period of interhemispheric coupling. Later in the text we will show that the shape given by Eq. (1) represents the diurnal cycle in clouds better than a simple harmonic fit, but prior to validation of the

shape one has to introduce a general diurnal cycle estimation approach itself. The examples shown in Fig. 2 utilize the $A(t)$ given by Eq. (1), though the approach will remain valid for any periodic function.

Figure 2 explains the approach for estimating the diurnal variation phase and amplitude: let us imagine that a real diurnal variation for a given type of cloud at a given location is defined by Eq. (1), with $A_{24}$ and $\varphi_{24}$ known (black curve in Fig. 2a).

5   For this case, the red circles in Fig. 2a correspond to the values obtained at the local observation times of AIRS and IASI, which are passed to the diurnal phase/amplitude estimation algorithm. To test the sensitivity of the approach to the uncertainties of the amplitudes related to uncertainties of AIRS and IASI cloud amounts, we also consider the case when a 20% random "noise" is added to the amplitudes at the sampled observation times.

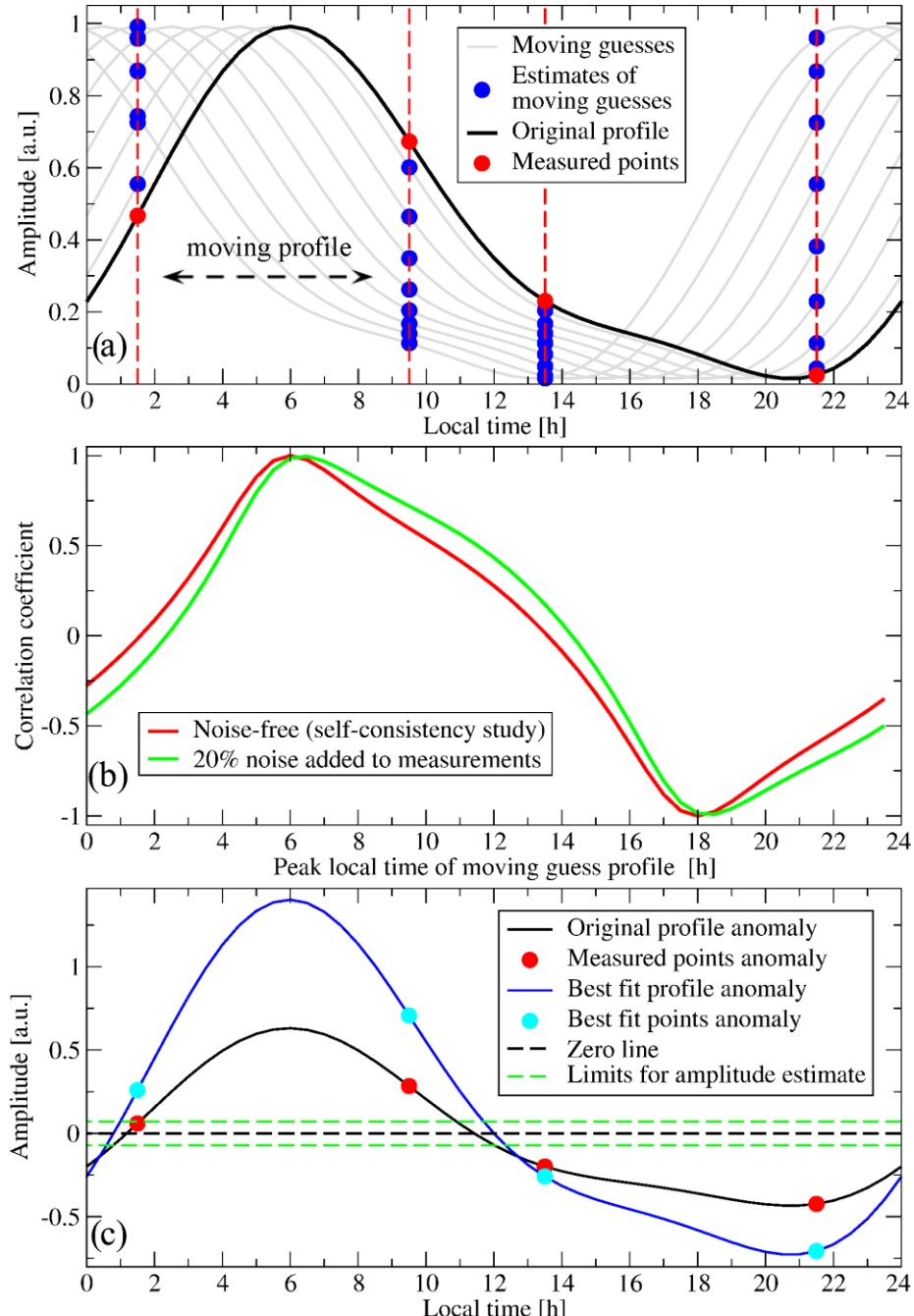

**Fig. 2. Illustration of the approach to estimate the diurnal variation phase and amplitude from four measurements, taking advantage of a known form of the variation: a) "True" profile measured at four points and moving guess profiles; b) Pearson's correlation coefficient calculated for guess profiles for noise-free and noisy simulations; c) determination of the amplitude with the phase known.**

5  This uncertainty was estimated from the most recent CIRS-AIRS and CIRS-IASI cloud products (Stubenrauch et al., 2017) which were compared with active lidar cloud measurements of the CALIPSO mission (Winker et al., 2009). The comparison

showed a "hit rate" for individual measurements in the tropics and midlatitudes of the order of 88% over oceans and of 82% over land (hit rate stands for the ratio of number of cases, for which AIRS cloud detection agrees with CALIOP, to the total number of cases).

The first step in the analysis is to build the function in accordance with Eq. (1) with an *arbitrary amplitude $A_{24}$*. Then we numerically solve the system for $\varphi_{24}$ as follows. The $\varphi_{24}$ is changed in fine increments, each time calculating *A(t)* at the four local observation times of AIRS and IASI (blue circles on grey curves in Fig. 2a). A set of obtained values is compared to a reference "measurement" (red circles), and a Pearson's correlation coefficient $k_{corr}$ is calculated for each phase shift (Fig. 2b, phase converted to peak time for the sake of visualization). In the noise-free self-consistency study the maximum of $k_{corr}$ should exactly match the phase reproducing the original function. The tests show that even 20% random noise added to the "reference" points does not spoil the phase determination by more than half of an hour. Since the peak of the $k_{corr}$ curve is not sharp, we make a conservative estimate of the uncertainty of our method of phase (or peak local time) determination to be ±1.5 h.

With the phase known, we estimate the $A_{24}$ amplitude (Fig. 2c) as follows: we draw a virtual "zero line" at the level corresponding to a mean of all four points and calculate the magnitudes at these points with respect to this "zero line". Then we compare the obtained values with those estimated from the Eq. (1) in the same way (dashed black line in Fig. 2c represents a zero line of Eq. 1). The mean ratio of amplitudes gives $A_{24}$. Since using the values close to "zero line" might lead to zero-over-zero type of errors and to an increase of the $A_{24}$ uncertainty, we pick up only the *A(t)* points with the amplitude greater than 0.2 of the maximal $|A(t)|$ value (the threshold is marked by green dashed lines in Fig. 2c). Another way of estimating $A_{24}$ is to compare the maximal spans of the reference profile sampled at four observation times and that of the measured one. We find these methods to be equivalent, but the one involving more points should be less noisy and, therefore, more reliable. The noise in the measured points affects the uncertainty on $A_{24}$, with 20% noise in the source data leading to about 20% uncertainty in the estimated amplitude.

To justify the choice of the relationship in Eq. (1) for the fitting, we have performed the following numerical experiment using real data: one year of AIRS and IASI cloud data using the methodology outlined above for two different hypotheses on the fitting functions: a simple harmonic one with a 24-h period and a mixture of diurnal and semi-diurnal described by Eq. (1). For each tested hypothesis, we have built a histogram of the best correlation coefficient values, separately for high- and low-level cloud amount diurnal variation. We found that using Eq. (1) for the fitting of real-life observations one achieves ~8% and ~18% higher correlation coefficients for the diurnal cycle of high- and low-level clouds, respectively, than with a simple harmonic function fitting. Searching a better fitting shape of the diurnal variation is out of the scope of this study, but the approach to estimate phase and amplitude of the diurnal variation, under the assumption of a known and fixed shape of the diurnal variation, remains valid for any periodic function (e. g. see the surface temperature variation fitting in Appendix B).

Summarizing this section, the "sliding profile approach" allows to estimate the phase and amplitude of the diurnal variation from four measurements per day performed at arbitrary time with respect to peak time. The uncertainty of the estimated peak time for the combination of four AIRS and IASI monthly averages over 1° latitude × 1° longitude is ±1.5h while the diurnal cycle amplitude is estimated with ~20% uncertainty.

## 3   Diurnal phase and amplitude of high clouds and their surrounding

### 3.1.   Zonal averages

We apply the diurnal cycle estimation algorithm on the amount of all high-level clouds and separately on high opaque, cirrus, and thin cirrus cloud amount from the CIRS-AIRS and CIRS-IASI cloud climatologies. For the following analyses we determined $A_{24}$ and $\varphi_{24}$ for each month and each 1º latitude ×1º longitude grid box, calculated $A(t)$ in accordance with Eq. 1, and averaged the resulting shapes for a given latitude band or region.

To demonstrate the feasibility of our methodology, we compare diurnal variations of high-level cloud amount averaged over three latitudinal bands, separately for ocean and land, to those presented by Noel et al.(2018), obtained from new lidar measurements of the Cloud-Aerosol Transport System (CATS, Palm et al., 2016, Yorks et al., 2016) aboard the International Space Station (ISS), and to those from ISCCP presented by Rossow and Schiffer (1999). While lidar observations are more sensitive to thin cirrus than IR sounders, ISCCP is less sensitive to thin cirrus (e. g. Stubenrauch et al., 2013). The much larger statistics of several years of global AIRS/IASI and ISCCP data compared to only nadir track statistics of three years of CATS leads to a smoother behaviour of the diurnal variations presented in Fig. 3 for boreal summer. The shape assumption in the diurnal cycle estimation used for AIRS/IASI also contributes to a smoother behaviour and is well compensating for the better temporal resolution of ISCCP. All three datasets show larger diurnal cycle amplitudes over land than over ocean, in agreement with many other studies (e. g. Soden, 2000; Tian et al., 2004; Stubenrauch et al., 2006; Zhang et al. 2008). They are the largest in the tropics, and towards higher latitudes they decrease in winter (see also Fig. S1 for Southern hemisphere in the supplement), as shown for AIRS/IASI.

Differences in the amplitudes between AIRS/IASI and CATS may be mostly explained by the small sampling of CATS, as only a few regions per latitude band are sampled each time. With increasing sampling towards higher latitudes (to be checked in the CATS paper), the agreement increases: over midlatitude land, both curves agree very well. ISCCP agrees in general also well with AIRS/IASI, in particular in the tropics and the subtropics, except during night where the high-level cloud amount is slightly underestimated by ISCCP due to misidentification of thin cirrus as midlevel cloud (Stubenrauch et al., 1999a). Over summer midlatitude land, the lower sensitivity of ISCCP to thin cirrus also leads to a slight diurnal amplitude underestimation and in addition to a slight phase shift (see also Fig. 4).

Concerning the phase, the agreement between AIRS/IASI and CATS and between AIRS/IASI and ISCCP is indicated in Fig. 3 by Pearson's linear correlation coefficients, together with the local peak times. The values are in general high, when the diurnal amplitudes are large (above noise), which is the case in the summer hemisphere and tropics over land. The peak local times

(marked in red and blue) of AIRS/IASI and CATS do agree to within 2 hours that is comparable to the estimated uncertainty range of 1.5 h. Concerning ISCCP, the local peak time in the tropics and subtropics has been systematically determined earlier, because of the slight underestimation of high-level cloud amount during night, while over summer midlatitude land the local peak time is estimated there hours later as by AIRS/IASI and CATS, because of missing thin cirrus during daytime. The latter

are responsible for the local peak time at 17h LT, as seen in Fig. 4 which presents the contributions of the different cloud types (high opaque, cirrus and thin cirrus) to the total diurnal variation of high clouds for the same latitude bands as in Fig. 3, during boreal summer in the Northern hemisphere (Fig. S2 for austral summer in the Southern hemisphere). Again, amplitudes are larger over land than over ocean, and in general the amplitudes of the individual cloud types are larger than of all high-level clouds mixed together, as the phases of these cloud types differ.

The distinction between the cloud types also allows a deeper interpretation of the diurnal cycle. The diurnal cycle over tropical land, which is already largest for all high-level clouds together, as illustrated in Fig. 3, can be mainly understood as the diurnal cycle of deep convective cloud systems, with a peak of precipitation around 18h (Tian et al., 2004; Zhang et al. 2008), followed by high opaque cloud amount, including convective cores as well as part of thick anvil cirrus, with a peak around 21h and developing cirrus anvil amount with a maximum around 1h LT. The diurnal cycle of thin cirrus is smaller than the one of

cirrus, because part of the thin cirrus corresponds to dissipating convective cloud systems and part to cirrus formed *in situ* by large-scale forcing (e. g. Luo and Rossow, 2004; Riihimaki et al., 2012). While the phase of cirrus and thin cirrus is lagged in the tropics, with a minimum of thin cirrus in the morning and a minimum of cirrus around 13h LT, their phase gets more similar towards higher latitudes. This can be probably explained by the fact that *in situ* freezing TTL cirrus do not exist at higher latitudes. In the midlatitudes cirrus and thin cirrus have a maximum in the afternoon and are most probably linked to

synoptic situations of fronts. Some cirrus may be orographic, generated by ascent of air within large amplitude vertically propagating waves over mountains and even over hills (e. g. Queney, 1948; Ludlam, 1952).When comparing tropical land with tropical ocean, we observe a difference of about 11 hours for the high opaque clouds, with a broader maximum amount occurring in midmorning, again a few hours later than convection and precipitation of early morning (e. g. Tian et al., 2004; Zipser et al., 2006; Zhang et al., 2008). Cirrus and thin cirrus follow with maxima in the evening and during night. Their

minima are again shifted, with the minimum of cirrus just before the maximum of high opaque clouds, as over land. The minimum of thin cirrus occurs in midmorning, similar to the minimum over land. The similar diurnal cycle of thin cirrus over ocean and over land is another indication that some of these thin cirrus are formed *in situ*, having no direct relation to the convective systems. The difference in phase of convection between ocean and land and the much broader peak of convection over ocean has been associated with differences in the vertical structure of land and ocean convection by Soden (2000). In

contrary to land, over open ocean the thermal properties of the ocean surface undergo a relatively weak diurnal cycle. Chen and Houze (1997) have shown that the life-cycle of convective cloud systems also plays a role in affecting the diurnal cycle of cloudiness. The formation of longer lived oceanic convective cloud systems may introduce a bi-diurnal cycle, with large systems occurring at the same location only every other day.

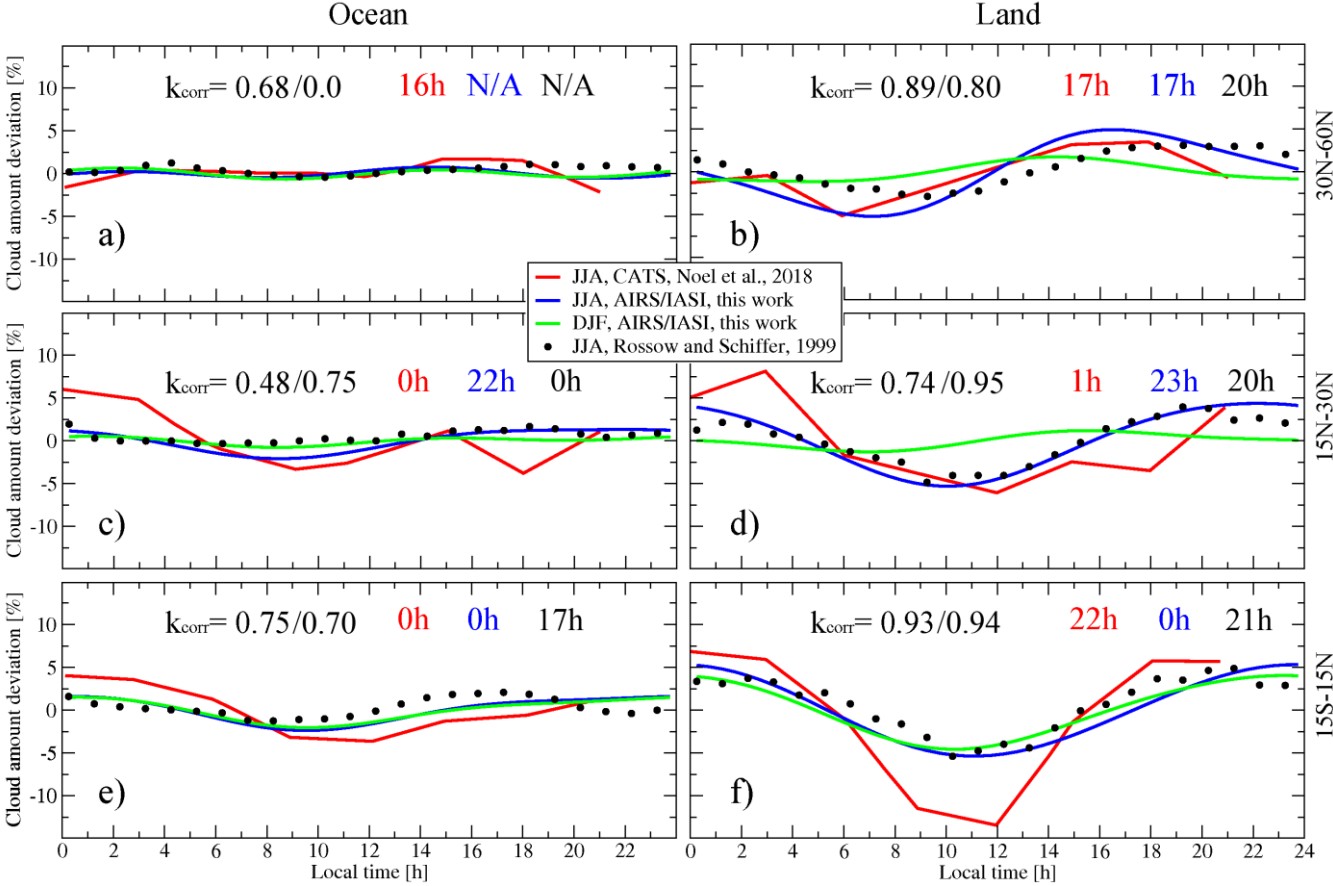

**Fig. 3.** Comparison of average diurnal cycles of high-level cloud amount over three latitudinal bands estimated from CATS lidar observations (Noel et al., 2018), from AIRS/IASI (this work), and from ISCCP (Rossow and Schiffer, 1999). CATS statistics only includes June, July, and August (JJA, red curves) whereas AIRS/IASI results are shown both for boreal summer (JJA, blue curves) and boreal winter (December, January and February, DJF, green curves) in 2008−2015. The ISCCP JJA data cover the period of 1989–1991. The correlation coefficients are given for AIRS/IASI vs CATS and AIRS/IASI vs ISCCP sequences for JJA in the upper left part of each panel; the peak local times for CATS are marked in red, for AIRS/IASI in blue, and for ISCCP in black in the upper right part of each panel. a) 30N−60N, ocean; b) 30N−60N, land; c) 15N−30N, ocean; d) 15N−30N, land; e) 15S−15N, ocean; f) 15S−15N, land.

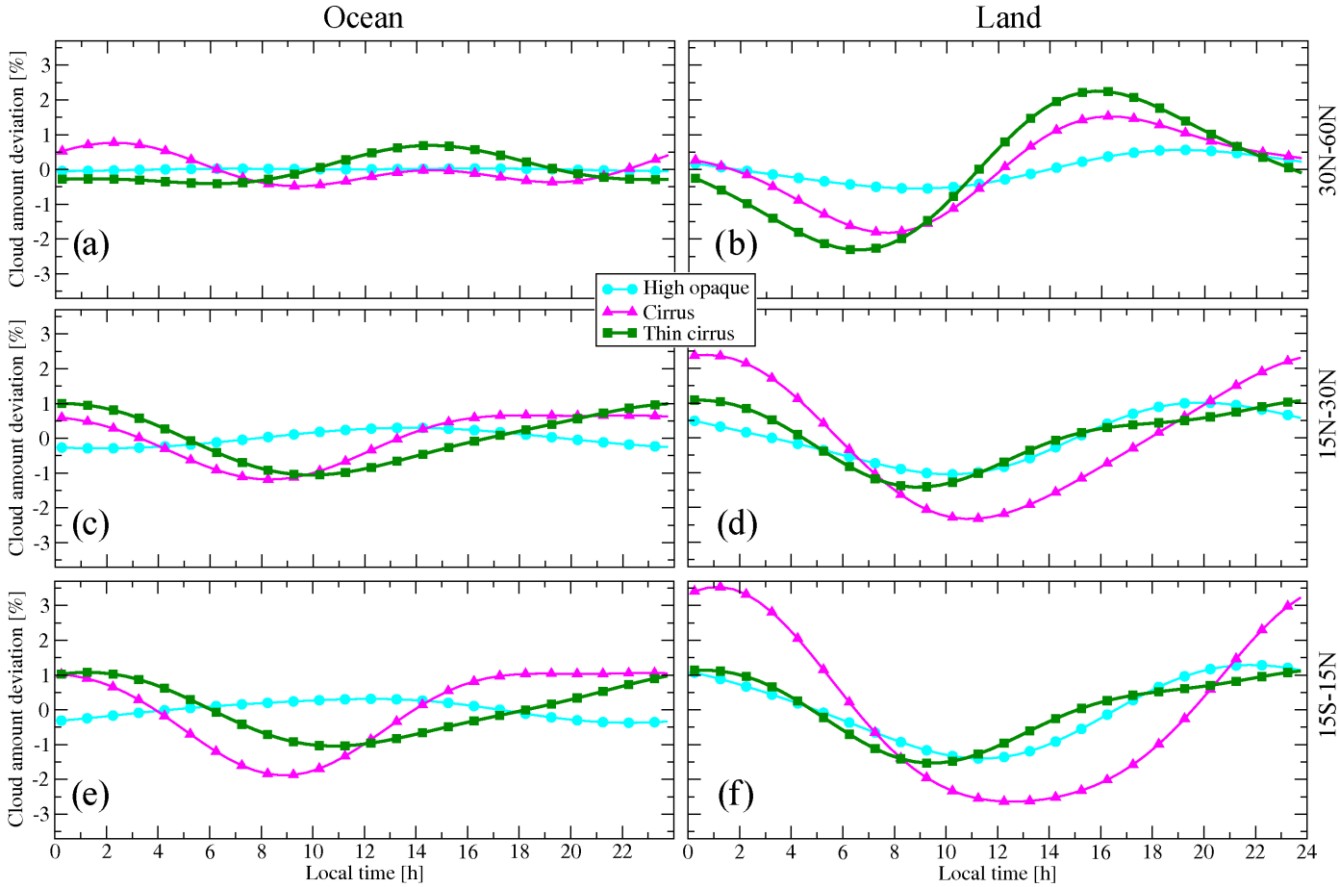

**Fig. 4.** Diurnal cycle of high opaque, cirrus, and thin cirrus amount in in NH midlatitudes, NH subtropics and tropics in boreal summer: a) 30N−60N, ocean; b) 30N−60N, land; c) 15N−30N, ocean; d) 15N−30N, land; e) 15S−15N, ocean; f) 15S−15N, land. The AIRS/IASI statistics is averaged for 2008−2015.

## 3.2.    Geographical distributions

We first present geographical maps of phase and amplitude of the diurnal cycle of the high-level cloud types and then explore the links to their atmospheric environment. The latter includes surface temperature ($T_{surf}$) and upper tropospheric (UT) relative humidity (RH), the parameters, which are both linked to their formation and then affected by the clouds.

We apply the methodology described in Section 2.2 to cloud type amount statistics gathered over grid cells of 1° latitude x 1° longitude for the period 2008 - 2015, separately for January and July. Average diurnal amplitude and phase are represented by vectors as suggested by Cairns (1995) and also utilized in (Soden, 2000; Tian et al., 2004), where the vector's length corresponds to the amplitude of the diurnal variation and the phase is converted to a local peak time, given as the direction of the vector. Since different cloud types are characterized by different diurnal amplitudes, a unit vector is added to the lower right corner of each panel. We consider the diurnal cycle to be reliably detected at a given latitude/longitude, if the Pearson's

correlation coefficient for the corresponding diurnal curves (see Section 2.2) is greater than 0.85, an empirical threshold based on examining numerous diurnal variation curves.

Whenever we average the diurnal cycle parameters, we calculate the mean phase (or peak time) using the corresponding amplitudes as weights. To avoid errors caused by averaging the phases in the vicinity of 24-0h transition (for example, direct
averaging of 23h and 01h returns noon instead of midnight), we apply a "resulting force" algorithm (Appendix A).

Figures 5 and 6 present detected diurnal variations for the three different high-level cloud type amounts, for January and July, respectively. A common feature of all these maps is that the amplitudes of the diurnal variations maximize over the tropical belt and in the summer hemisphere midlatitudes. This is an expected behaviour consistent with other observations (e. g. Rossow and Schiffer, 1999; Wylie and Woolf, 2002; Tian et al., 2004; Hong et al., 2006; Stubenrauch et al., 2006). Figs.5 and 6 of
(Hong et al., 2006) present peak times of precipitation, of very cold cloud amount (IR brightness temperature $T_B^{IR} < 210$ K) and of cold cloud amount ($T_B^{IR} < 235$ K) for different tropical regions, with lags between the three, corresponding to the development of deep convective systems. Compared to these results, our results are consistent when associating high opaque clouds with very cold clouds and cirrus with the warmer clouds. Another expected feature is the magnitude of diurnal change over land being generally larger than that over ocean, as the ocean surface temperature has a much smaller diurnal cycle
(Fig. B1 in the Appendix B).

In general, the high opaque cloud amount with about 5% is much smaller than the cirrus and thin cirrus amount (Fig. 1). The clouds identified by the CIRS retrieval as high opaque ones, for which a diurnal variation is detected, have an average diurnal amplitude of about 5%, but certain regions (Fig. 5a, 6a) demonstrate amplitudes reaching 10%. High opaque clouds, often associated with deep convective cores in the tropics, have a large regional variability, which may be influenced by orography.
Maximum high opaque cloud amount moves with the ITCZ towards the summer hemisphere, with peak times between 18h and midnight. In July, there is a large contrast between continental opaque clouds and nearby oceanic regions. Over the continents, the peak typically often occurs in the evening around 20h as compared with oceanic areas near the continents with peaks closer to noon. Nonetheless, in some other ocean locations opaque clouds also peak in the evening or overnight (e.g. tropical longitudes –115 to –135). The diurnal variation of cirrus clouds (Fig. 5b, 6b) is larger than that of high opaque clouds:
the average amplitude is ~8%, with some regions reaching the values of 12% and individual grid cells showing amplitudes of up to 20%. There is less contrast between land and ocean for the cirrus and thin cirrus peak times: with continental peak times between midnight and 2h and oceanic ones around 18h. Over land, a lag of about 3 hours can be identified for two thirds of the cases (see also Section 3.3). It seems to be more complicated to identify the lags over ocean, with a maximum of cirrus in the evening. Thin cirrus have a maximum at midnight over ocean, lagging behind cirrus, which can be interpreted as the
thinning of convective systems during night. Over land the situation is a bit more complicated, with a maximum after noon in large mountain areas (Rocky Mountains, South America, Africa and Asia) and some lagging behind cirrus, again to be interpreted as thinning of convective systems. UT relative humidity, presented in Fig. 7, seems to be maximum in the convective regions just a few hours after the maximum of cirrus and thin cirrus, as observed by (Horvath and Soden, 2008).

While relatively small cloud amounts (like the one of high opaque clouds) or relatively cloud similar amounts during the day (like over ocean) provide a "noisier" input for the diurnal cycle estimation, large "blank" areas in the winter hemisphere with no detected diurnal variation assure us that the algorithm is stable against false triggering provoked by noise, and the diurnal cycle of cirrus clouds detected over the tropical ocean in July (Fig. 6b) is very close to that reported in (Soden, 2000) for their averaged June-August, 1987 (10%, 18h). This allows to say that larger ocean zones, which demonstrate a consistent diurnal cycle, deserve attention. However, the global maps have been established with one single detection threshold using Pearson's correlation coefficient of 0.85, so if one wants to focus on specific ocean regions one might want to refine the threshold on the correlation coefficient.

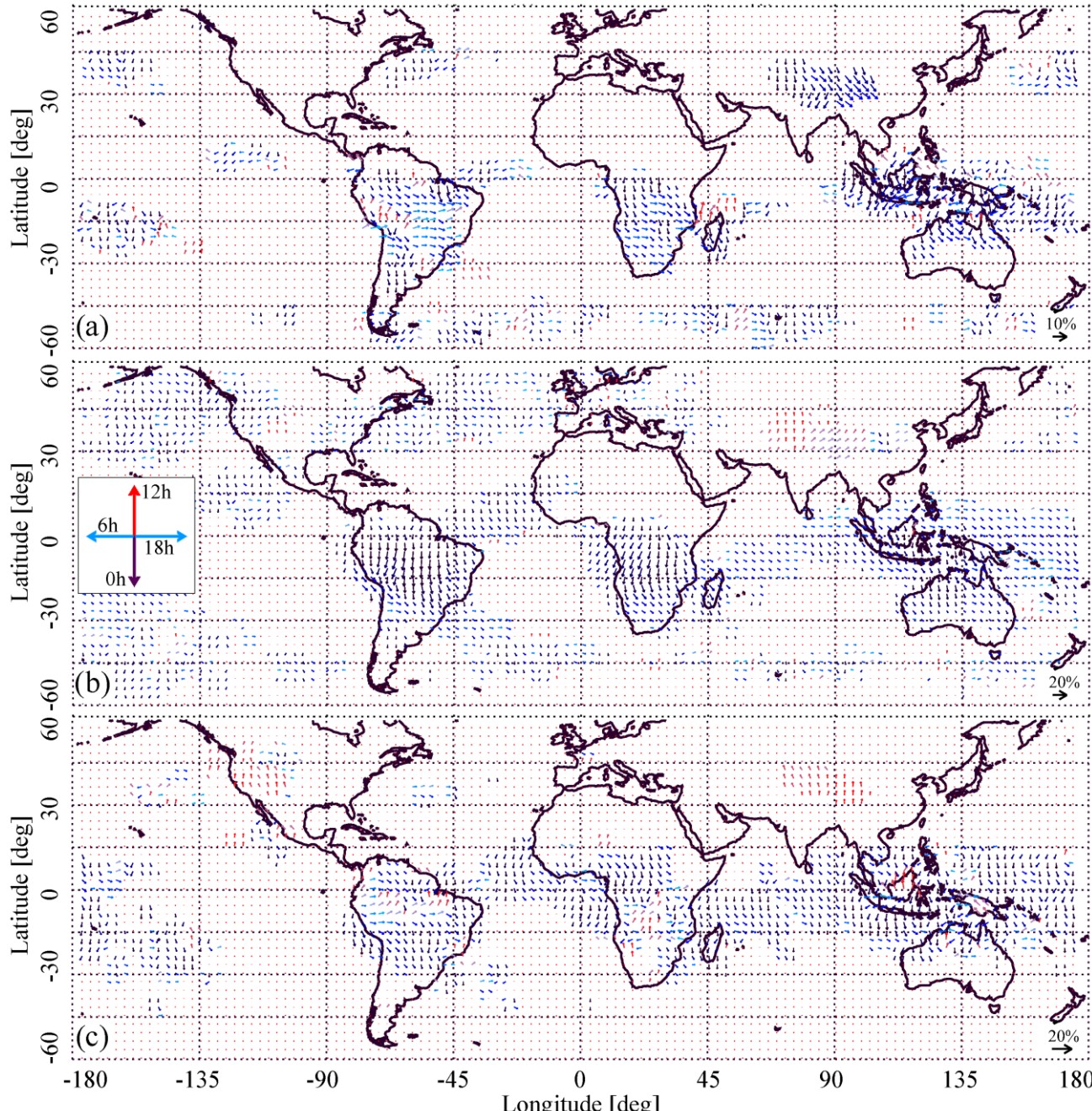

Fig. 5. Diurnal variation of (a) high opaque clouds, (b) cirrus clouds, and (c) thin cirrus clouds estimated from AIRS/IASI for January. Statistics is averaged over the period of 2008–2015. Vector length is proportional to the amplitude and its orientation defines the local time of the peak: downwards for 0h, left for 6h, upwards for 12h, and right for 18h. For the sake of readability, the arrows are also coloured in tints of red for day 6h−18h and in tints of blue for night 18h−6h.

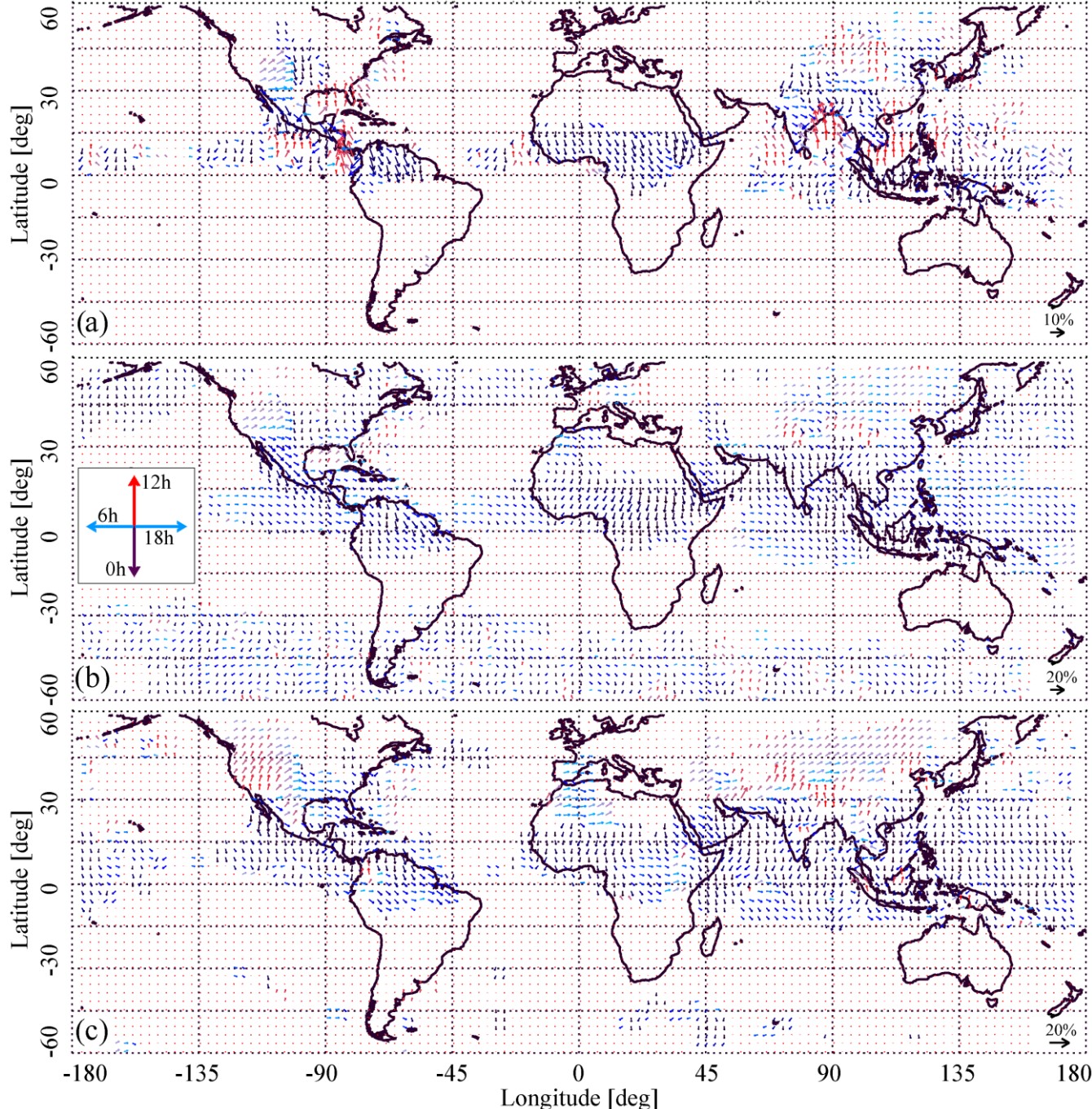

**Fig. 6. Same as Fig. 5, but for averaged July (2008 through 2015).**

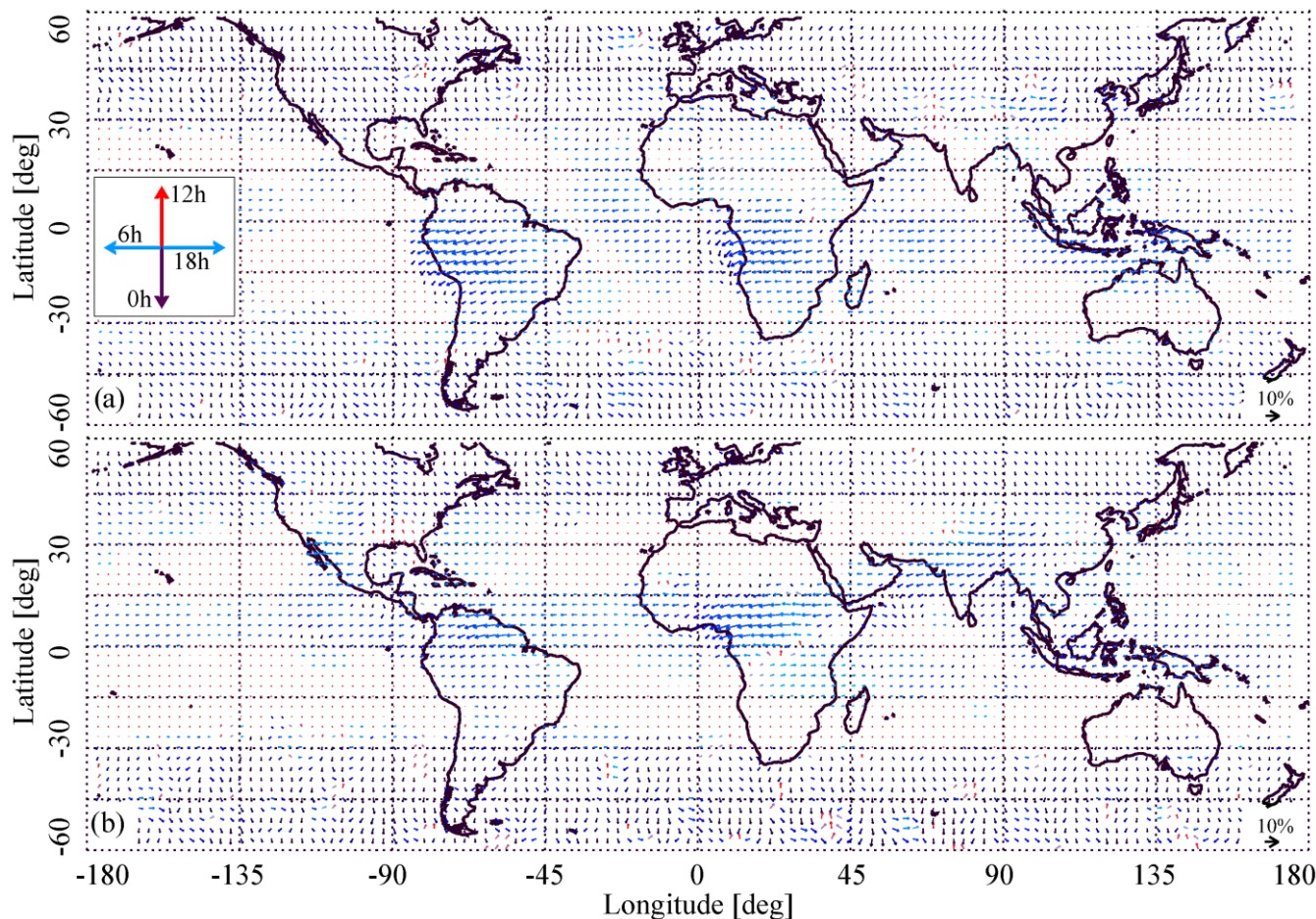

**Fig. 7. Diurnal variation of average relative humidity in a layer 150 hPa below the tropopause, estimated from ERA-Interim reanalyses, a) for January and b) for July. The vector representation is consistent with the caption of Fig. 5. The tropopause level was determined in accordance with (Reichler et al., 2003).**

## 3.3.     Specific regions

Since different geographical regions are characterized by different cloud regimes (e. g. Rossow et al., 2002), we define seven regions over land, presented in Fig. 8, which we analyse more in detail. Regions 1, 2, 3 and 4 are in the Northern midlatitudes and subtropics / tropics, while regions 5, 6 and 7 are in the Southern subtropics / tropics. The regional amplitudes and peak times of cloud type amount, $T_{surf}$ and UT relative humidity are summarized in Tables 1 and 2 and illustrated in Fig. 9 which presents circular histograms of the peak amplitude local times, separately for January and July.

From these tables we deduce that the two regions in the Northern midlatitudes (1 and 2) show a large difference in the $T_{surf}$ amplitude, with a larger one in summer. The regions which are affected by the ITCZ (4, 5 and 6) have a slightly larger $T_{surf}$ amplitude in summer, while the two regions in the subtropics (3 and 7) show a large $T_{surf}$ amplitude both in January and in

July. In general, a strong convective activity is revealed by a very large diurnal cycle in the occurrence of high opaque clouds (more than 10%). This is the case for regions 6 and 7 in January and for regions 3 and 4 in July.

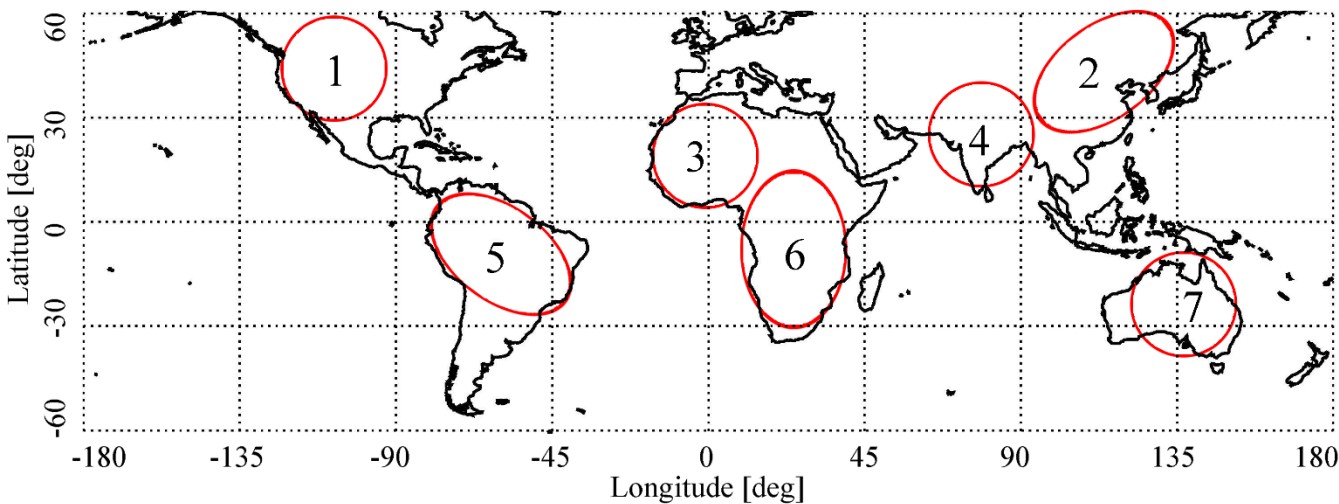

**Fig. 8. Regions selected for a more detailed analysis of the cloud diurnal cycle over land: 1 – North America, 2 – North/East Asia, 3 – Northern Africa, 4 – Indian continent, 5 – South America, 6 – South Africa, 7 – Australia.**

The large diurnal cycle in $T_{surf}$ in summer (about 10 K) of the two regions in the Northern midlatitudes (1 and 2) leads to some convective activity (diurnal cycle of high opaque clouds of about 7% and 5%, respectively) in the afternoon. However, multiple peak amplitude local times may indicate effects of orography. The signals for cirrus and especially thin cirrus are more evident, with similar peak amplitudes and more concise peak amplitude local times in the early afternoon. The thin cirrus may be orographic cirrus. UT relative humidity has two peaks, one in the early morning (5h) and one in the afternoon (17h), in both cases the afternoon peak lagging shortly behind cirrus and thin cirrus and the early morning peak lagging behind high opaque clouds during night. During winter, the diurnal $T_{surf}$ amplitude is much smaller (about 5 K), and no diurnal cycle could be detected in high opaque clouds. The diurnal cycle in UT relative humidity has opposite peaks, again at 5h and 17h. Over North America, cirrus has multiple peaks while thin cirrus has a peak around 13h and a smaller one in the morning (7h), while over North-East Asia there is one peak detected in cirrus around 14h. It is difficult to relate the UT humidity clearly to the cirrus during winter.

The two other regions in the Northern hemisphere are subtropical (3) and tropical (4). While the diurnal $T_{surf}$ cycle is large both in January and in July (11 K) over North-West Africa, it is large (10 K) during the summer monsoon over the Indian subcontinent, while it is smaller (4 K) in January, when the ITCZ moves southwards. The diurnal behaviour of UT clouds is quite different between these regions, whereas the peak amplitude times in UT relative humidity are similar (17h in winter and 5h in summer). In the subtropics, in summer there seems to be some convective activity with a peak time in the evening (21h – 24h), followed by the development of cirrus and thin cirrus during night (2h), whereas in winter no diurnal cycle of high opaque clouds could be detected, and cirrus and especially thin cirrus have multiple peak times. The summer monsoon over the Indian subcontinent leads to a peak amplitude time of high opaque clouds in the early evening, indicating convective

activity, followed by the development of cirrus anvils in the night and thin cirrus until noon. In January, the amplitude of the diurnal cycle of high opaque clouds is only half and later during night, while thin cirrus develop already in the morning, thickening towards cirrus in the afternoon.

| variable / zone | January | | | | | July | | | | |
|---|---|---|---|---|---|---|---|---|---|---|
| | $T_{surf}$ | Opq. | Thick | RH | Thin | $T_{surf}$ | Opq. | Thick | RH | Thin |
| N. America | | N/D | | | | | | | | |
| N.-E. Asia | | N/D | | | N/D | | | | | |
| N.-W. Africa | | N/D | | | | | | | | |
| Indian cont. | | | | | | | | | | |
| S. America | | | | | | | | | | |
| Africa | | | | | | | | | | |
| Australia | | | | | | | N/D | N/D | N/D | N/D |

**Fig. 9. Circular histograms of local time of peak amplitude for $T_{surf}$, high opaque clouds, thick cirrus, relative humidity (RH), and thin cirrus for average January and July. The convention for vector direction representation is consistent with that described in the caption of Fig. 6 while the amplitude is proportional to normalized histogram count number. "N/D" marks the cases, for which the diurnal cycle could not be reliably detected.**

Comparing the tropical regions in the Southern hemisphere, South America has a slightly smaller diurnal amplitude of $T_{surf}$ than the African region (5 K compared to 7 K), as the latter also includes subtropical land and the former is part of the 'Amazonian green ocean'. In summer (January), maximum convective activity in South America seems to be from 15h on, with cirrus and thin cirrus during night. Thin cirrus also has another peak amplitude in parallel with high opaque cloud between noon and 18h. Over Africa the diurnal behaviour is similar. In winter (July), the peak amplitude times of high opaque clouds are again later, with cirrus during night and again thin cirrus at multiple times.

Table 1. Diurnal cycle amplitude and local time of the peak for $T_{surf}$, high opaque clouds, thick cirrus, thin cirrus, and relative humidity for average January. Long dash symbol (−) means that no reliable cycle was detected while a star symbol (*) corresponds to multiple peaks in the circular histogram (see Fig. 9)

| Variable | $T_{surf}$ | | High opaque | | Cirrus | | UT Relative humidity | | Thin cirrus | |
|---|---|---|---|---|---|---|---|---|---|---|
| Geographic area | Ampl. [K] | Peak [h] | Ampl. [%] | Peak [h] | Ampl. [%] | Peak [h] | Ampl. [%] | Peak [h] | Ampl. [%] | Peak [h] |
| N. America | 4.6 | 12.3 | − | − | 5.8 | 6-12 | 3.8 | 5; 17 | 5.2 | 7−13 |
| N. E. Asia | 6.3 | 14.0 | − | − | 7.4 | 15.1 | 3.3 | 5; 17 | − | − |
| N. W. Africa | 11.0 | 12.3 | − | − | 5.9 | − | 2.2 | 16.4 | 5.2 | * |
| Indian cont. | 9.7 | 12.0 | 5.7 | 22−2 | 5.3 | 15.2 | 1.8 | 17.9 | 4.8 | 9−15 |
| S. America | 4.9 | 12.5 | 7.5 | 16.1 | 12.8 | 23.3 | 5.2 | 5.0 | 7.3 | 13−19 |
| Africa | 6.9 | 12.7 | 7.9 | 17-22 | 12.2 | 1.3 | 4.4 | 4.8 | 7.8 | * |
| Australia | 11.3 | 12.1 | 7.5 | 20.3 | 8.2 | 19; 1 | 2.0 | 5.5 | 6.2 | 15.5 |

Table 2. Same as Table 1, but for average July.

| Variable | $T_{surf}$ | | High opaque | | Cirrus | | UT Relative humidity | | Thin cirrus | |
|---|---|---|---|---|---|---|---|---|---|---|
| Geographic area | Ampl. [K] | Peak [h] | Ampl. [%] | Peak [h] | Ampl. [%] | Peak [h] | Ampl. [%] | Peak [h] | Ampl. [%] | Peak [h] |
| N. America | 10.5 | 11.8 | 7.2 | 14-2 | 7.7 | 16-24 | 2.9 | 5; 17 | 7.5 | 14.8 |
| N. E. Asia | 8.3 | 12.0 | 5.2 | * | 6.8 | 15.7 | 2.8 | 5; 17 | 6.4 | 15.9 |
| N. W. Africa | 11.0 | 12.5 | 6.9 | 22.7 | 8.1 | 1.4 | 2.7 | 4.9 | 7.3 | 2.3 |
| Indian cont. | 4.3 | 11.9 | 8.3 | 19.5 | 11.7 | 22.7 | 4 | 5.3 | 8.0 | * |
| S. America | 5.6 | 13.2 | 8.1 | 1.0 | 8.8 | 21.2 | 2.9 | 5.1 | 6.3 | 18.9 |
| Africa | 7.8 | 13.4 | 7.7 | 22.1 | 11.2 | 1.2 | 3.5 | 5; 17 | 7.0 | * |
| Australia | 8.5 | 12.6 | − | − | − | − | 1.5 | 5; 17 | − | − |

For Australia, a diurnal cycle was only detectable in summer (January), with two peaks in high opaque clouds and in cirrus (19h and 1h) and a peak in thin cirrus in the early afternoon.

In general, UT relative humidity has a peak in the early morning for regions of convection, which may be explained by UT humidification from the dissipation of the anvils (e. g. Horvath and Soden, 2008). In winter, there seems to be a peak in UT
10  relative humidity in the afternoon, often also just after the appearance of thin cirrus, which this time seem to be formed *in situ*, and not being offspring of a convective system.

For the regions where diurnal variation was reliably detected both for high opaque clouds and for cirrus, the diurnal cycle amplitudes of these two cloud types are correlated with $k_{corr}=0.75$. This can be probably assigned to being part of the same cloud system. Even though peak times for cirrus clouds and for high opaque clouds vary almost in the same range, an average lag of ~3 h can be identified for two thirds of the cases. We have to note that in certain cases the definition of the lag becomes ambiguous due to possible 24 h phase shift, which is not detectable in our approach. For example, the 15 h peak in January histogram for thick clouds over South Asia (Fig. 9) can be caused both by *in situ* formation of high clouds 4 hours after the peak of local insolation and by an outflow of the high opaque column with a characteristic time of ~16 h (the lag between high opaque and cirrus cloud peaks).

Another presumable element of the cloud system lifecycle is the dissipation of the anvil. This can be manifested both in the RH change (Fig. 7) and in thin cirrus variation (Fig. 5c, 6c). The peak time of RH "release" from the cloud oscillates between late afternoon and early morning maxima (Kottayil et al., 2016) and its lag with respect to cirrus can be estimated as ~3.5 h. Finally, thin cirrus show good coupling with high opaque clouds and the cirrus, and the amplitudes of their diurnal cycle are comparable with those of the cirrus. The peak times for these clouds have broad distributions that makes it difficult to estimate the lags. We assign this to long characteristic times of anvil dissipation and to mixing the effects of *in situ* and convective outflow cloud formation mechanisms: air parcels saturated with water vapour released in the process of anvil dissipation may travel to other areas and form clouds there. For the cases when the peak time distribution is narrow, the average lag with respect to anvil can be estimated as ~10 h, but the individual values vary from almost zero to 20 h.

These lags and correlations indicate that the convective cloud life cycle might be described as follows: (a) the convective cloud peak time precedes the cirrus anvil formation; (b) the cirrus anvil dissipates, releasing water vapour and turning to thin cirrus; (c) both the cirrus anvil and thin cirrus are strongly coupled with the high opaque core; (d) relative humidity is strongly coupled with the cirrus, lagging behind which may be associated with upper tropospheric humidification by cirrus outflow.

### 3.4. Relating diurnal amplitudes to climate fluctuations

As the period of eight years of combined AIRS and IASI observations is too short to directly study long-term variability of diurnal phases of the different cloud type amounts, we present in this section geographical patterns of diurnal amplitude variability in relation to climate variability, given by de-seasonalized monthly mean global surface temperature, $T_{surf}$, anomalies. Within this short time period, changes in global $T_{surf}$ (or tropical, as both are strongly correlated) reflect the El Niño Southern Oscillation (ENSO). ENSO is the most dominant mode of interannual variability in the Earth's climate system, and has been often used to study cloud feedbacks (e. g. Lloyd et al., 2012; Liu et al., 2017; Stephens et al., 2018). In general, a positive global $T_{surf}$ anomaly corresponds to El Niño, with maximum convection over the Central Pacific and a negative $T_{surf}$ anomaly to La Niña, with maximum convection over the West Pacific. Stubenrauch et al. (2017) have shown with 15 years of AIRS-CIRS cloud data that changes in relative amount of tropical high opaque and thin cirrus with respect to increasing global mean $T_{surf}$ have different geographical patterns (see their Fig. 12): while the high opaque clouds, often linked to strong

precipitation (Protopapadaki et al., 2017), relative to all clouds, increase in a narrow band in the tropics, there is a large increase in relative thin cirrus amount around these regions.

By applying the same technique, namely by determining a change in diurnal amplitude of cloud type amount as a function of change in global mean $T_{surf}$ by a linear regression of their monthly time anomalies, at a spatial resolution of 1°latitude x
1°longitude, we have obtained the spatial patterns presented in Fig. 10. Indeed, we observe also an ENSO pattern for the change in diurnal amplitude, with an increasing amplitude in regions of deep convection (the band near the equator, in the Central Pacific for El Niño and for the West Pacific and South America for La Niña). The strongest changes are observed for high opaque and cirrus cloud amount, but one can distinguish them also for the thin cirrus which seem to be directly linked to convective cloud systems. Considering the relative errors of the determined slopes, presented also as geographical maps in the
supplement (Fig. S3), though in general quite noisy, we see nevertheless that the strongest signal over the Central Pacific has the smallest error (of about 30%). With this analysis we have shown yet in another way that high-level clouds linked to deep convective systems have the largest diurnal amplitudes. To study the change in phase is more delicate, already due to possible misinterpretation of peak amplitude around midnight and also as other factors such as winds may play a role.

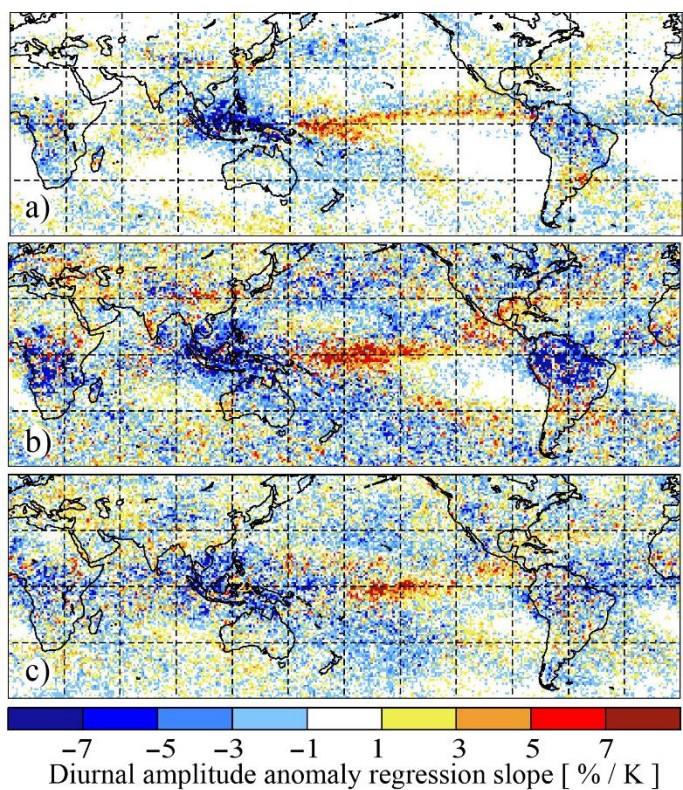

**Fig. 10. Geographical maps of linear regression slopes for the period of 2008–2015 estimated from the ratio of de-seasonalized monthly mean diurnal amplitude anomalies determined from combined AIRS-CIRS and IASI-CIRS cloud data, and global mean $T_{surf}$ anomalies, from ERA-Interim for (a) high opaque cloud amount, (b) cirrus, and (c) thin cirrus.**

## 4. Conclusions

Multi-spectral infrared sounders are advantageous for the retrieval of the high-level cloud properties. Their good spectral resolution allows a reliable cirrus identification down to an IR optical depth of 0.1, day and night. However, these instruments are mostly aboard polar orbiting satellites, providing only observations twice per day. In this article, we presented a methodology to use the synergy of AIRS and IASI cloud observations to address the diurnal variation not only of total high-level cloud amount, but also separately for high opaque, cirrus, and thin cirrus clouds.

Based on previous studies (Stubenrauch et al., 2017), we needed to implement the same set of surface and atmospheric ancillary data (from meteorological reanalysis ERA-Interim) to extract a reliable diurnal cycle from the cloud retrievals performed on different satellite instrument measurements. We demonstrated the feasibility to determine the diurnal cycle amplitude and phase from just four measurements per day using the "sliding profile approach", which is based on the correlation of a measured variation with an assumed shape of the diurnal cycle. For the combination of AIRS and IASI, this approach allows estimating the diurnal variation phase with an accuracy of ±1.5h while the amplitude is determined with ~20% accuracy.

The zonally averaged diurnal cycle of high-level cloud amount estimated from AIRS and IASI cloud data compares relatively well with other datasets, such as the one determined from CATS lidar observations and the one from ISCCP multispectral imager observations. Slight differences can be understood by much more limited statistics of CATS, in particular in the tropics, and misidentification of thin cirrus by ISCCP.

Considering diurnal variations separately of high opaque, cirrus and thin cirrus amount leads to a better understanding, as one can also study the lags between the different cloud types, which also have different radiative effects. In general the amplitude of the diurnal cycle is larger over land than over ocean and largest in the tropics, followed by summer midlatitudes, in agreement with other analyses. By using the time series of the eight years of combined AIRS-CIRS and IASI-CIRS data, we could relate diurnal amplitude changes to climate fluctuations, given by de-seasonalized global $T_{surf}$ anomalies from ERA-Interim, and found that largest diurnal amplitudes of high-level cloud amount are linked to deep convective systems.

A more detailed analysis of specific regions over land has shown that the largest diurnal cycle seems to prevail during summer monsoon over the Indian subcontinent with 8.3% for high opaque clouds, 11.7% for cirrus, and 8.0% for thin cirrus clouds. The local peak times vary with cloud type, season, and location, but in general AIRS alone with its observations at 1h30 and 13h30 LT is closer to capturing the maximum and minimum of cloud cover over tropical land than IASI observing the atmosphere at 9h30 and 21h30 LT. Lags and correlation coefficients between high opaque, cirrus, thin cirrus and UT relative humidity indicate a life cycle of continental tropical convective systems as: (a) the convective cloud peak time precedes the cirrus anvil formation; (b) the cirrus anvil dissipates, releasing water vapour and turning to thin cirrus; (c) both the cirrus anvils and thin cirrus are strongly coupled with the high opaque core.

**Data availability**

The monthly database of detected diurnal cycle amplitude and phase for UT clouds (high opaque, cirrus and thin cirrus), at a spatial resolution of 1° latitude x 1° longitude, from the AIRS-IASI synergy can be downloaded from the ResearchGate repository using the following information: Feofilov, A., and Stubenrauch, C.: Diurnal cycle of high clouds retrieved from the

synergy of AIRS and IASI infrared sounders for 2008-2015, ResearchGate, DOI: 10.13140/RG.2.2.13038.15681, 2019.

**Author contribution**

Artem Feofilov performed the cloud property retrievals from AIRS and IASI observations, developed the methodology of diurnal variation retrieval from the combination of two infrared sounders, and analysed the data. Claudia Stubenrauch compared the retrieved diurnal variations with those retrieved from CATS lidar, performed the zonal analysis and established

the links between the cloud system elements.

**Competing interests**

There are no competing interests for this article.

**Acknowledgements**

This research was supported by ESRIN (Contract No.: 4000101773/10/I-LG) within the framework of ESA Climate Change

Initiative (ESA CCI), Phase I and by CNRS.

## Appendix A:    Resulting force algorithm

To avoid error caused by averaging the phases in the vicinity of 24-0h transition (for example, direct averaging of 23h and 01h returns noon instead of midnight), we apply a "resulting force" algorithm, which is resembling the calculation of the tilt of a disk with masses on its edges (Fig. A1): the positions of the "masses" on the disk's perimeter correspond to phase values while the masses themselves are proportional to the amplitudes. When everything is set up and the disk is "released", the direction of the tilt defines the average phase (arrow in Fig. A1b).

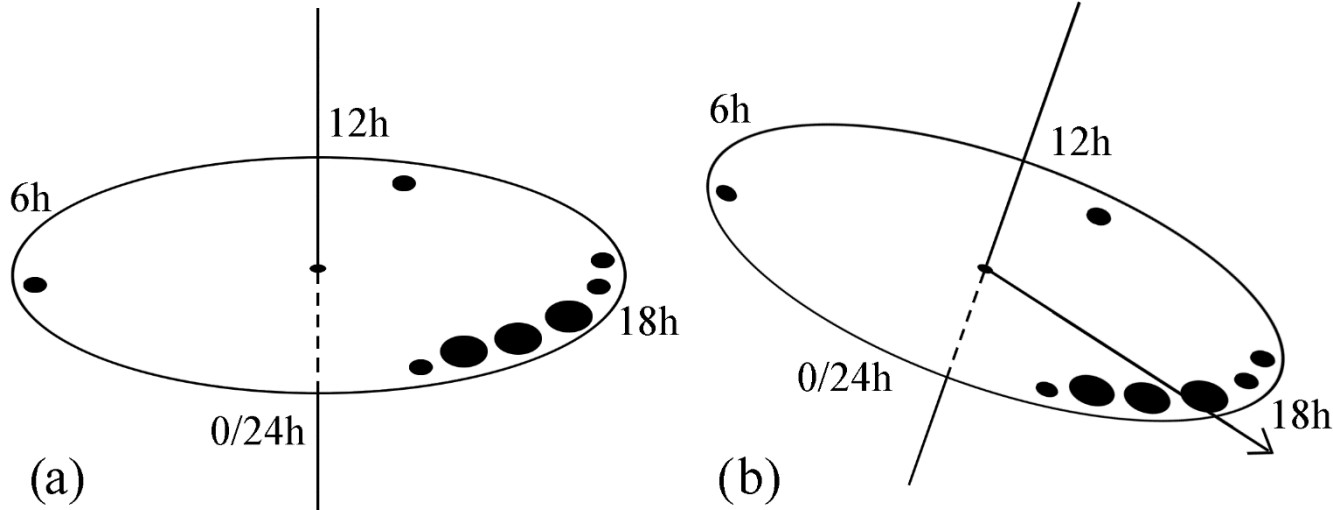

**Fig. A1. Illustration of "resulting force" algorithm for calculating the average of phase values: a) diurnal cycle amplitudes are used as "weights" (black dots of different sizes), which are placed on the perimeter of the disk in accordance with the corresponding phase values; b) when "released", the disk tilts in the direction of the most frequent phases with the strongest amplitudes.**

## Appendix B: Diurnal variation of surface temperature

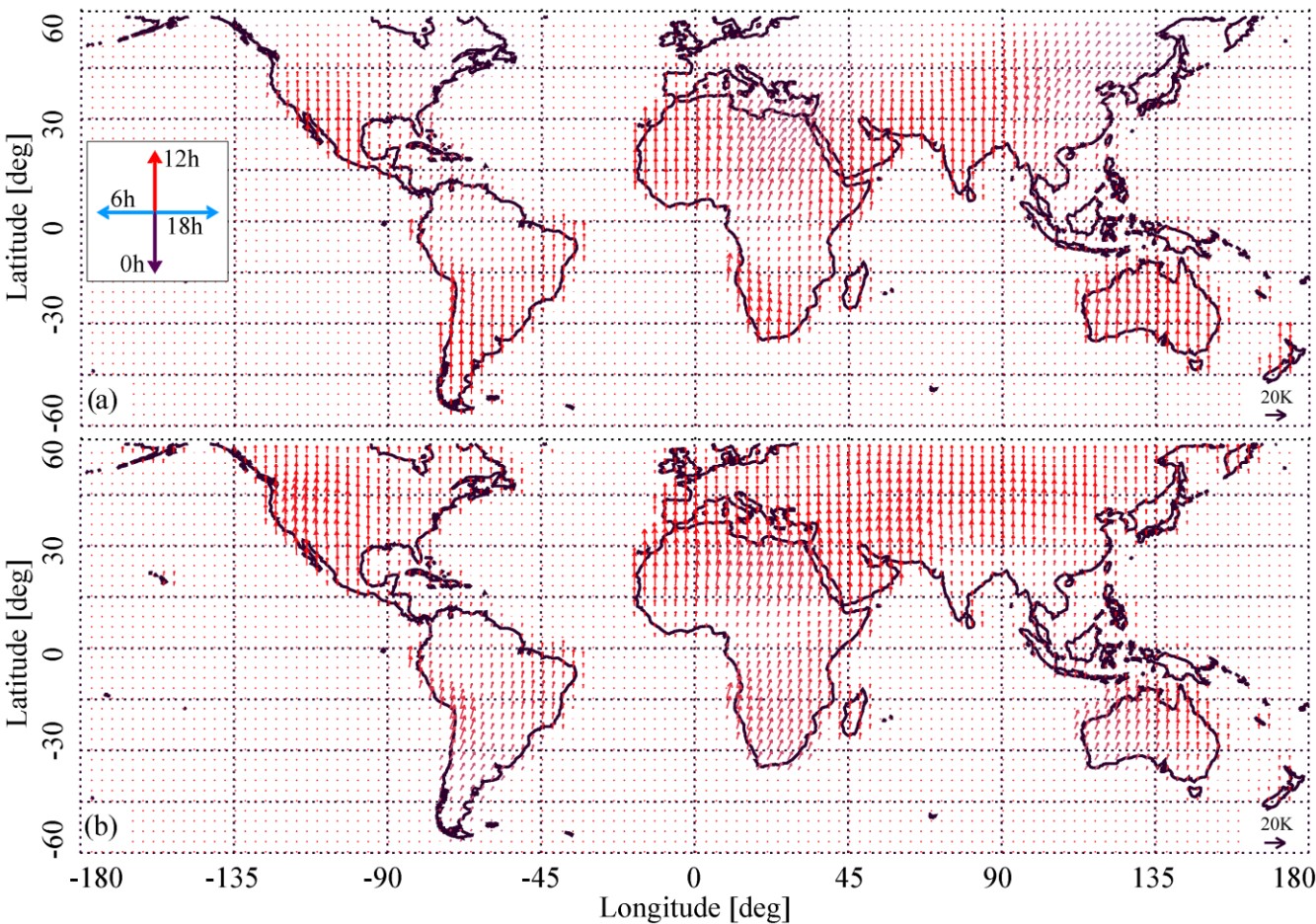

**Fig. B1.** Diurnal variation of surface temperatures, estimated from ERA-Interim reanalyses, in January (top) and in July (bottom). The vector representation is consistent with the caption of Fig. 5.

5   We searched the parameters of diurnal variation of surface temperature using the shape provided by (Aires et al., 2004), which we approximate as follows:

$$A(t) = 0.95 \cdot \left[ \frac{1}{t^2+15} + e^{-0.08 \cdot (t-13.2)^2} + 0.04 \cdot t \right] \qquad (B.1)$$

where the amplitude $A(t)$ is in [K] and time $t$ is in [h] and the coefficients come from the best fit approximation. As one can see (Fig. B1 and Tables 1 and 2), the amplitude of $T_{surf}$ variation over land is large in tropical areas and in the summer

10  hemisphere, reaching 20 K in the deserts (the doubled amplitude corresponds to $max(T_{surf}) - min(T_{surf})$ temperature span) while the variation over ocean is mostly negligible. The absolute values of the $T_{surf}$ variation agree with those reported in (Goetsche and Olesen, 2001; Pinker et al., 2007; Duan et al., 2014; Holmes et al., 2015; Ruzmaikin et al., 2017). The spreading of the diurnal cycle from coastal regions out to surrounding oceans has been noted already in (Yang and Slingo,

2000) who suggested a complex land-sea-breeze effects as an explanation. The local time of the peak is quite stable in all areas where the diurnal cycle was detected, both in January and July, and we estimate the interannual variability for the $T_{surf}$ peak time for each zone to be ~0.5 h. Depending on the geographical area, $T_{surf}$ peak time changes within 11.8−14.0h limits, and possible mechanisms of the lag with respect to peak of solar insulation are discussed in (Ait-Mesbach et al.2015).

According to their simulations, the soil thermal inertia "impacts directly the amplitude of $T_{surf}$ variation with lower thermal inertia inducing higher amplitude (and vice versa)". Their study shows that thermal inertia also impacts the turbulent heat fluxes.

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
