# Peer review of "Diurnal variation of high-level clouds from the synergy of AIRS and IASI space-borne infrared sounders"

_Atmospheric Chemistry and Physics, 2019_

## Referee Comment (RC1) · Anonymous Referee #2 · 23 Apr 2019

In this paper, the authors analyze global-scale cloud detections derived from measurements by the AIRS and IASI instruments, to infer the diurnal cycles of high-level clouds over oceans and several land regions in the tropics.

The paper is well-written and structured. It is easy to follow and makes its arguments convincingly. The methods used to analyse the data appear sound and described with detail. The evidence is well presented and supports the paper's conclusions. The results are useful and relevant to the field. I appreciate the way the authors derived a daily cycle from a couple of measurements made at two local times, and the focus on well-selected land regions. I have no problem recommending the paper for publication,

although I have a few very minor comments below.

**Minor comments**

- The first two paragraphs of the introduction lack a few references.

- p. 3, l. 21: CIRS has already been described and Stubenrauch et al., 2017 already cited higher on the same page (l. 1 and 2, respectively), please fix

- p. 5, l. 19: "the phase shift... was found": by who? Is this part of your results?

- p. 5, l. 24: the "moving profile" approach is quite smart, did the authors invent it? If so please state it, otherwise provide a reference to previous use

- p. 7, l. 25: "This justifies using Eq. (1) for the analysis." The experiment described between l. 20 and 25 justifies using Eq. (1) for the analysis *over the 24-h harmonic function selected by the authors*. It does not proves that Eq. (1) is the best function to use for the analysis. For instance, it would be possible to conjure many additional functions which might prove a worse fit than Eq. (1), but it would still not justify Eq. (1) as the best choice for the analysis. I understand the authors explain that searching for a better function is beyond the scope of the paper in the following sentence, and I'm fine with that, but the statement above is still incorrect. Unless I have misunderstood, please revisit the reasoning of this paragraph and make it more robust.

- section 3 : High clouds are identified unambiguously in CATS data by the altitude from which the lidar signal is backscattered to the instrument. This is not the case for the cloud detections documented in the AIRS/IASI dataset. Could you comment on how the uncertainties in cloud altitude in the AIRS/IASI dataset

might affect the retrieved diurnal cycle of high clouds in one way or another, and if these effects are consistent with the differences with CATS results?

- section 3, figure 3: the comparison with CATS is interesting, but how do the AIRS/IASI cycles compare with the ISCCP daily cycles described by Rossow and Schiffer 1999? Their results are presented as part of the introduction, why not compare them to the AIRS/IASI results in addition to CATS in Fig. 3?

- p. 11, l.9: Wyley -> Wylie

- Fig. 6: A more direct legend would be "Same as Fig. 5 for July"

- p. 18, l. 10: "(Fig. 8)" -> Fig. 9? (Fig. 8 shows the land regions)

- p. 19, l.17: Maybe a dumb question, but who wrote the paper?

- p. 19, l.26: Where were the CATS data shown in Fig. 3 obtained?

- p. 24, l.30: I tried getting the Wylie and Woolf paper by following the doi link but it does not work. Please fix it

---

## Referee Comment (RC2) · Anonymous Referee #3 · 17 Jul 2019

Review of Manuscript: acp-2019-166

Title:

Diurnal variation of high-level clouds from the synergy of AIRS and IASI space-borne infrared sounders

Authors: Artem G. Feofilov and Claudia J. Stubenrauch

Overview:

In this article, Foefilov and Stubenrauch present a nice analysis of the diurnal cycle of high clouds as derived from AIRS and IASI space-borne infrared sounders. While I think the description of the analysis technique could be clearer and the article would benefit from more concrete discussion on how the results are similar (or not) to previous studies, my only serious concern is that the article does not have sufficient discussion of uncertainties due to assumptions made in the retrieval approach or uncertainties do to sampling limitations.

While I hesitate to call such a "major revision" (as it may well be possible to fix these shortcomings with relatively minor changes to the text), I nonetheless think it is important that these uncertainties be addressed in a quantitative way (much as the 20% uncertainty in the diurnal amplitude due to random error is established).

Recommendation: Major Revisions

General Comments:

1) Description of fitting technique / section 2.2.

At first reading, I found the material in section 2.2 difficult to follow. In particular, it was not initially clear to me that you were fixing BOTH the ratio A12/A24 and delta_phi. At least that is what I think you are doing.

Question 1A) Is delta_phi for all your high cloud analysis fixed to 0?

Question 1B) Is this value of 0 also based on Cairns 2005? (If yes please make this clear in the text, and otherwise explain how this value is determined).

Assuming the answer to 1A above is yes, I suggest you modify the text (starting somewhere around page 5 line 20) to read:

"With the A12/A24 set to 0.28 and delta_phi set to 0, the diurnal "shape" of Eq. (1) is fixed (see the gray line in Fig. 2a) and the problem reduces to one of determining the amplitude A24 and phi_24. There are four total measurement (measurements at four times of the day) and the problem is therefore overdetermined. Thus we

determine the best fit amplitude and phase using a "sliding profile approach", as depicted in Fig 2 and described below.

As regards the description of Figure 2 and the fitting process:

Page 7, line 13-15.  I don't follow this description.  Please rephrase.  Why should a cloud fraction value of 0.2 yield a zero over zero?  Does this mean the first red point in Fig 2c (near 1 UTC) is not used in the amplitude calculation?

It is in no sense obvious to me that your sliding profile approach is any better than doing a least squares fit or any other minimization approach.   Not that I think that it needs to be better, per se, but if you have a specific rationale for developing this approach rather than using established techniques for over determined problems it would be good to explain such.

**2) Uncertainty in diurnal amplitude and phase due to ASSUME A12/A24 ratio and delta_phi.

You assess the uncertainty in your results due to the impact of a 20% random uncertainty in the cloud fraction, the result of which is a 20% uncertainty in the amplitude (which is not surprising) and 1.5 hours uncertainty in the phase.  This is a GOOD result!  But what about the assumed values for A12/A24 and delta_phi?  I can easily imagine that even a small change in either of these assumed values might have a large impact. In particular, I expect these values might have spatially coherent regional variations that might impact your later analysis.

**3) Uncertainty due to sampling variability.

I am very surprised to see that you do not discuss the sampling variability.  Is the sampling variability small relative to the diurnal variations?   For example, if you plotted the same result using any randomly chosen 5 or 8 years of data (rather than all 10 or 15 years) would all the results be so close together that I couldn't see any differences in any of the plots or values presented in Tables 1 and 2?

4) What is new here ?

Do the results of the diurnal analysis differ from results of prior observational studies or extend our understanding of the diurnal cycle in some way?   While reproducibility of results is an important aspect of science and having some "new result" is not strictly necessary, I think the article would be more interesting if it does.   In general, more concrete discussion on how the results are similar (or not) to other studies would make the analysis more valuable.

Minor Comments:

Introduction, first paragraph:  I know this is well known material, but if you are going to start off a publication with such, you aught to support the material with appropriate references.   Frankly, I think you might just drop this first paragraph and start the introduction with the second paragraph "Due to the importance of clouds ... " (line 3, page 2).

Page 2, line 5.   Starting phrase is awkward, I suggest change to read simply, "The diurnal variation modulates ...".

Page 2, line 8.  Perhaps change to "Many early global observational analyses ... ".  I thought Rossow had a few papers looking at diurnal cycles before 1995 (1983 perhaps ???) and I think there we some earlier studies yet (pre ISCCP).  My point I is the there is no need to declare "first", and in general this might depend on how one defines "global".

Page 2. Line 10. Perhaps change "are significant" to "were found to be".

Page 2.  Line 20-23.   I think these three bullet points are also intended to apply to cloud over land in the tropics – and don't for example apply to mid-latitude in the winter.   Perhaps move the phrase "over land in the tropics" from the first bullet into line 18, or otherwise clarify.

Page 4, line 1. What does "transformed" mean here?

Section 2.1.  The spatial resolution of AIRS and IASI is coarse compared with most IR imagers.   Perhaps it would be good to include some comments on how resolution impacts the cloud classes: high opaque ($\varepsilon$cld > 0.95), cirrus (0.95 > $\varepsilon$cld > 0.5), and thin cirrus (0.5 > $\varepsilon$cld > 0.1).

Page 4, line 1.  Also what does the correction to SST entail?    If it is truely unimportant to the study results, then perhaps leave this line out.   If it does matters, then you should probably explain a bit more.

Page 4, line 5.  Perhaps use "pixels" or "fields of view" rather than "scenes".   To me "scene" implies a collection of pixels on some scale, for example a 60 km scene.   As such a scene might have high-cloud even if only a fraction of the pixels contain clouds.

Page 5, line 6.   The word "admixture" seems a bit archaic to me and in general the idea of the diurnal cycle being a mixture of semi-dirunal cycles seems odd.   Perhaps change to "... since variations in cloud amounts are known to include variations on both diurnal and semi-diurnal time scales."

Page 7, line 1.  What do you mean by "hit rate"?   How does this translate into a 20% random noise?

Page 7, line 3. Perhaps change "build the function" to "solve for A23 and phi_24". Also as far as I can see, you are only looking at high clouds in this study, so delta_phi is simply 0.

Page 7, line 4. Perhaps change to "Having solved for phi_24, ..."

Page 7, line 13-15. I don't follow this description. Please rephrase. Why should a cloud fraction value of 0.2 yield a zero over zero?

Page 8, line 3. Change "... separately of high opaque ..." to "... for high opaque ...".

Figure 3. What is being "averaged" in Figure 3? Are the data being spatially averaged (if so what was the starting spatial scale)? Or temporally averaged (what time period)? For example do you first calculate the diurnal amplitude and peak time for each 1x1 latitude region (for each cloud type) using all years of data and then average spatially, or using 1 year of data and average in space and time ?

Page 8, line 14. Early studies. Please provide specific references.

Page 8, line 20. I presume you mean with the estimate 1.5 hours uncertainty. Perhaps clarify. (As a very minor point I note that 0 h to 22 h is a 2 hour difference, so perhaps just say, they agree to within 2 hours).

** Top of Page 9. What does "In situ Freezing" mean?

** Page 8 / figure 4 Discussion.

A) What about mid-latitudes? Don't they deserve some discussion? In particular, (A1) Over land, why are cirrus and thin cirrus cloud in phase and LEAD opaque cloud? This is very different than tropical/subtropical land. (A2) Over mid-latitude land, diurnal variation of thin cirrus is larger than cirrus while the opposite is true for subtropics and tropics (where cirrus larger than thin cirrus). This is worth a comment, I think. (C3) Over mid-latitude ocean, are the relatively small variations real (larger than uncertainty)? If yes, does this mean that cirrus form more prominently overnight and thin during the day Or might this be a retrieval artifact or ??

B) Subtropics ocean is similar to tropical ocean and subtropical land is similar to tropical land for boreal summer. This doesn't seem too surprising. What about boreal winter? Are subtropics similar to tropics in boreal winter? Similar to mid-latitudes? Or differ from both? (Perhaps put NH winter and SH winter plots in supplementary material).

C) In general, is there anything here that is new or different from previous studies?

Page 11, line 2.  0.85 seems an arbitrary choice.  I presume your results are NOT sensitive to this choice, meaning if you choose 0.7 it has little impact ?  Please comment.

Page 11, line 10.  Are the differences with Hong … Random?  Systematic is some regions?  Earlier or later?

Page 11, line 22.  On stability.  I agree that there are many fewer points with a retrieved diurnal cycle in the winter hemisphere but there are some signals that perhaps deserve some comments.  In Figure 5a, is the feature in the (A) Atlantic off the US East Coast, (B) the Central Pacific, and (C) that Himalayas real?   (Do other climatologies show these features?  Any idea what is going on here?).  Likewise, I wonder if the diurnal cycle observed of the Southern Ocean is real.

Page 11, line 26.  Perhaps change to "In July, there is a large contrast between continental opaque clouds and nearby oceanic regions.   Over the continents, the peak typically often occurs in the evening around 20h as compared with oceanic areas near the continents with peaks closer to noon.  Nonetheless, in some other ocean locations opaque clouds also peak in the evening or overnight (e.g. tropical longitudes -115 to -135)."

Page 11, line 27.  As regards Indonesia, I think the Maritime continent has a complex land/sea breeze interactions that I don't think the 1 degree data can capture well.  I suggest it might be better to simply note this (and perhaps cite appropriate references) rather than trying to make any conclusions for this region.

Figure 8.   I suggest changing the numbering scheme used in Figure 8 and the discussion to match the order of the regions shown in Figure 9 and tables #1 and #2 from top to bottom.   This will make the text easier to follow.

Section 3.3.  What guided the selection of regions analyzed here?  In particular, why are no oceanic regions selected for analysis?  Also, the North America region that is highlighted includes the west coast and Rockies but misses most of the central U.S. and Eastern U.S.  Was the intent to avoid Mesoscale Convective Systems (MCSs) that strongly influence the diurnal phase and amplitude in the Centeral/Southern US?   I note that Figure 6a shows the Central and Eastern US is notably different from the western US.

Page 15, line 8. I agree that multiple peaks could be explained by orographic effects, however, I also think that MCSs likely play some role here.   In general, what leads you to make this statement?  Is the intent that this statement is a conclusion Or are you speculating?

Figure 9.   Suggest change column header "Conv." To "Opq" in keeping with the figure caption, and to be consistent with the point that Opaque is not perfectly synonymous with convection.

Page 17, line 10.   As with earlier comment, I do not know what "in situ freezing" means.

Page 18, line 3.  I am not clear on what "in the same limits" means here.  Please rephrase.

Page 19, line 4-6.   You write, "It is interesting to note that the local time of the minimum moves from the summer hemisphere midlatitudes towards the tropics from 6h to noon and the one of the maximum from 17h to 1h."   I am confused, what is moving, the total amount of upper level cloud?   What does "one of the maximum" mean?  Perhaps expand and/or rephrase this remark.

Page 19, line 10.   What other analyses?  Suggest references.   Is there anything in the present work which adds to OR departs from the "other analysis?

---

## Author Comment (AC1) · 31 Aug 2019

**Response to Reviewer 1**
We thank the Reviewer 1 for his/her positive assessment of the manuscript and for the corrections. Below, we provide point-by-point answers to each of the comments.
We mark the reviewers' comments/questions and the authors comments/responses by "**RC:**" and "**AC:**", respectively.
Where it was appropriate, we modified the text of the manuscript in accordance with the recommendations. We provide a separate file to track the changes.

[Figure]

**General comments**

**RC:** The first two paragraphs of the introduction lack a few references.
**AC:** We have rewritten the introduction.

**RC:** p. 3, l. 21: CIRS has already been described and Stubenrauch et al., 2017 already cited higher on the same page (l. 1 and 2, respectively), please fix
**AC:** Perhaps, the reviewer was misled by the same name, but CIRS cloud climatology described in the beginning of the page is not the same as the CIRS cloud property retrieval package discussed in this line and (Stubenrauch et al., 2017) refers to both of them, so the reference should remain.

**RC:** p. 5, l. 19: "the phase shift... was found": by who? Is this part of your results?
**AC:** Yes, we parameterized the curves of (Cairns, 1995). We changed the text to "By analysing the plots of (Cairns 1995) using least-square fitting of Eq. 1 we found that the phase shift is equal to $-2$ h for the low- and mid level clouds and 0 h for high-level clouds"

**RC:** p. 5, l. 24: the "moving profile" approach is quite smart, did the authors invent it? If so please state it, otherwise provide a reference to previous use.
**AC:** We did not perform a search in literature for this approach and we cannot state that this is our "invention". We added a reference to (Goldberg et al., 2013) where a similar approach was used by one of the authors to find the period and phase of the interhemispheric coupling.

**RC:** p. 7, l. 25: "This justifies using Eq. (1) for the analysis." The experiment described between l. 20 and 25 justifies using Eq. (1) for the analysis over the 24-h harmonic

function selected by the authors. It does not prove that Eq. (1) is the best function to use for the analysis. For instance, it would be possible to conjure many additional functions which might prove a worse fit than Eq. (1), but it would still not justify Eq. (1) as the best choice for the analysis. I understand the authors explain that searching for a better function is beyond the scope of the paper in the following sentence, and I'm fine with that, but the statement above is still incorrect. Unless I have misunderstood, please revisit the reasoning of this paragraph and make it more robust.

**AC:** We'd like to refer the Reviewer 1 to the first figure we provided in the answers to the questions of Reviewer 2, which illustrates the aforementioned experiment performed using *real data*, and not for the *function selected by us* (the latter would give a perfect fit with all correlation coefficients equal to 1). As one can see, if the clouds exist (right-hand side of the plot), the "natural noise" related to uncertainties of the ancillary data and cloud retrieval methodology as well as to errors of parameterization given by Eq.1 leads to cases, for which the correlation coefficient is lower than 1. Still, the "semidiurnal fit" which corresponds to Eq. 1 gives higher correlation coefficients than a simple harmonic fit. The smoothness of the "tail" of the histogram and the general considerations regarding noise tell us that any other fitting function will not give a perfect fit, either and we are already close to good fitting.

To avoid the confusion, we rewrote the introductory sentence: "...we have performed the following numerical experiment using real data: one year of cloud data retrieved from AIRS and IASI with the help of CIRS has been processed..."

**RC:** section 3 : High clouds are identified unambiguously in CATS data by the altitude from which the lidar signal is backscattered to the instrument. This is not the case for the cloud detections documented in the AIRS/IASI dataset. Could you comment on how the uncertainties in cloud altitude in the AIRS/IASI dataset might affect the retrieved diurnal cycle of high clouds in one way or another, and if these effects are consistent with the differences with CATS results?

**AC:** This is a good question and it requires several references to be answered. In Fig.

2 of (Feofilov and Stubenrauch, 2017) we provide the basics for the cloud pressure error estimate in chi2-retrievals. As one can see, for high clouds, due to larger contrast between the radiation of the cloud and that of clear sky scene, the chi2-curve is steep and the pressure error is small. Correspondingly, the emissivity error is small, too (emissivity curve is estimated in $P\pm\Delta P$ points). In Fig. S3 of (Stubenrauch et al., 2017), the cloud heights retrieved from AIRS are compared with those from CALIOP and they show quite a good agreement and stability over the whole range of cloud emissivity. Finally, in the figures attached to this answer we show the emissivity error and pressure error distributions for high clouds. As one can see, both errors are small and therefore they should not affect high cloud determination accuracy to a significant degree.

**RC:** section 3, figure 3: the comparison with CATS is interesting, but how do the AIRS/IASI cycles compare with the ISCCP daily cycles described by Rossow and Schiffer 1999? Their results are presented as part of the introduction, why not compare them to the AIRS/IASI results in addition to CATS in Fig. 3?
**AC:** We followed the advice and added the curves of (Rossow and Schiffer, 1999) to Fig. 3 and modified the text correspondingly. The behavior of all three curves over land is quite consistent whereas the ocean areas with their weak amplitudes show moderate agreement. Please, also see the discussion around Fig.3.

**RC:** p. 11, l.9: Wyley -> Wylie
**AC:** Fixed, thanks.

**RC:** Fig. 6: A more direct legend would be "Same as Fig. 5 for July"
**AC:** We have changed the legend following the advice.

**RC:** p. 18, l. 10: "(Fig. 8)" -> Fig. 9? (Fig. 8 shows the land regions)
**AC:** Actually, we meant Fig. 7, relative humidity, but anyway thanks for pointing out this inconsistency.

**RC:** p. 19, l.17: Maybe a dumb question, but who wrote the paper?
**AC:** Originally, the draft of the paper was written by the first author, but then the paper converged to its final form in the course of several iterations, so now it is difficult to assign this or that part of the text to this or that author.

**RC:** p. 19, l.26: Where were the CATS data shown in Fig. 3 obtained?
**AC:** We took the data directly from the plots of (Noel et al., 2018) by digitizing them with the help of freely distributed Tracer 2.01 software by Marcus Karolewski (https://sites.google.com/site/kalypsosimulation/Home/data-analysis-software-1). In manual mode, the accuracy of digitizing is one screen pixel that is about 0.1

**RC:** p. 24, l.30: I tried getting the Wylie and Woolf paper by following the doi link but it does not work. Please fix it.
**AC:** The official doi copy-pasted from the journal's Web-site looks as follows: https://doi.org/10.1175/1520-0493(2002)130<0171:TDCOUT>2.0.CO;2 and if clicked it opens the page with the article (checked on the 31/08/19). Perhaps, a space or some other symbol spoiling the hyperlink was introduced on the author's or ACPD' side while formatting the file for online publishing. We believe, this will be checked and fixed by the production team when it comes to publication, but anyway, we've updated the link in the text.

Please also note the supplement to this comment:
https://www.atmos-chem-phys-discuss.net/acp-2019-166/acp-2019-166-AC1-supplement.pdf

[Figure]

[Figure]

**Fig. 1.** Left: Emissivity error histogram for high clouds; right: Pressure error histogram for high clouds

**Supplement:**

**Diurnal variation of high-level clouds from the synergy of AIRS and IASI space-borne infrared sounders**

Artem G. Feofilov and Claudia J. Stubenrauch

LMD/IPSL, Sorbonne Université, UPMC Univ Paris 06, CNRS, École polytechnique, Palaiseau, 91128, France

5 *Correspondence to*: A. G. Feofilov (artem. feofilov@lmd. polytechnique. fr)

**Abstract.** By covering about 30% of the Earth and by exerting a strong greenhouse effect, high-level clouds play an important role in the energy balance of our planet. Their warming and cooling effects within the atmosphere strongly depend on their emissivity. The combination of cloud data  from two space-borne
10 infrared sounders, the Atmospheric InfraRed Sounder, AIRS, and the Infrared Atmospheric Sounding Interferometer, IASI, which observe the Earth at four local times per day, allows us to investigate the diurnal variation of these high-level clouds by distinguishing between high opaque, cirrus, and thin cirrus clouds. We demonstrate that the diurnal phase and amplitude of high-level clouds can be estimated from these measurements with an uncertainty of 1.5 h and 20%, respectively. By applying the developed methodology to AIRS and IASI cloud observations for the period of 2008−2015, we
15 obtained monthly geographical distributions of diurnal phase and amplitude at a spatial resolution of 1° latitude x 1° longitude. In agreement with other studies, the diurnal cycle of high-level clouds is the largest over land in the tropics. At higher latitudes, their diurnal cycle is the largest during the summer. For selected continental regions we found diurnal amplitudes of cloud amount of about 7 % for high opaque clouds and 7 % for thin cirrus . Over ocean, these values are 2 to 3 times smaller. The diurnal
20 cycle of tropical thin cirrus seems to be similar over land and over ocean, with a minimum in the morning (9h LT) and a maximum during night (1h LT). Tropical high opaque clouds have a maximum in the evening (21h LT over land), a few hours after the peak of convective rain. This lag can be explained by the fact that this cloud type not only includes the convective cores, but also part of the thicker anvils. Tropical cirrus  show a maximum amount during night (1h LT over land). This lag indicates that they are part of the deep convective cloud
25 systems. However, the peak local times also vary regionally. We are providing a global monthly database of detected diurnal cycle amplitude and phase for each of these three high-level cloud types.

**1 Introduction**

30

~~are transparent in the shortwave (SW) and opaque in the long-wave (LW), allowing solar radiation to warm the surface and trapping the outgoing LW radiation, acting as a "greenhouse film". In addition to these counteracting processes, all clouds emit thermal radiation in all directions in accordance with their temperature, and the radiation escaping the atmosphere cools the planet. The algebraic sign of the net radiative effect of the cloud depends on its height, optical depth, vertical cloud layering, surface albedo and temperature, and local solar time.~~

[revised manuscript text omitted]
. We explain the difference between the diurnal cycle amplitudes shown in Fig.3 by (i) differences in geographical coverage (AIRS/IASI combination provides daily snapshots of the whole globe, whereas CATS needs about a month to cover the same area) and (ii) differences in sensitivity to optically thin clouds.
20  Fig.4 of 1.5 h. Concerning ISCCP, the local peak time in the tropics and subtropics has been systematically determined earlier, because of the slight underestimation of high-level cloud amount during night, while over summer midlatitude land the local peak time is estimated there hours later as by AIRS/IASI and CATS, because of missing thin cirrus during daytime. The latter are responsible for the local peak time at 17h LT, as seen in Fig. 4 which presents the contributions of the different cloud types (high opaque, cirrus and thin cirrus) to the total diurnal variation of high clouds for the same latitude bands as in Fig. 3, during
25  boreal summer in the Northern hemisphere (Fig. S2 for austral summer in the Southern hemisphere). BothAgain, amplitudes are larger over land than over ocean, and in general the amplitudes and of the individual cloud types are larger than of all high-level clouds mixed together, as the phases of these cloud types differ. This helps to explain the differences between the diurnal peak times obtained from instruments with different sensitivity to thin cirrus. The diurnal cycle will be shifted in phase accordingly.
30  The distinction between the cloud types also allows a deeper interpretation of the diurnal cycle: cirrus which are the most abundant in the tropics have the largest amplitude in the diurnal cycle, with a minimum around 1PM and a maximum around 1AM over land. High opaque clouds have a maximum at 9PM which is several hours later than convective precipitation in the afternoon. This can be explained by the fact that high opaque clouds include part of the thicker anvil which develop afterwards. After 9PM cirrus anvils continue to develop. Over ocean convection often occurs in the early morning (e. g. Zipser et al., 2006;

[revised manuscript text omitted]

---

## Author Comment (AC2) · 31 Aug 2019

We thank the Reviewer 2 for his/her thorough analysis of the manuscript and for the corrections.
Below, we provide point-by-point answers to each of the comments.
We mark the reviewers' comments/questions and the authors comments/responses by "**RC:**" and "**AC:**", respectively.
Where it was appropriate, we modified the text of the manuscript in accordance with the recommendations. We provide a separate file to track the changes (please see        https://www.atmos-chem-phys-discuss.net/acp-2019-166/acp-2019-166-AC1-

supplement.pdf)

**General comments**

**RC:** 1) Description of fitting technique / section 2.2
Question 1A) Is $\Delta\phi$ for all your high cloud analysis fixed to 0?
**AC:** Yes, it is. We added the sentence clarifying this in the text right after the Eq. 1.

**RC:** Is this value of 0 also based on Cairns 1995? (If yes please make this clear in the text, and otherwise explain how this value is determined)
**AC:** Yes, both values come from the figures of Cairns, 1995. We added an explanation to the text.

**RC:** Assuming the answer to 1A above is yes, I suggest you modify the text (starting somewhere around page 5 line 20) to read: "With the $A_{12}/A_{24}$ set to 0.28 and $\Delta\phi$ set to 0, the diurnal "shape" of Eq. (1) is fixed (see the gray line in Fig. 2a) and the problem reduces to one of determining the amplitude $A_{24}$ and $\phi_{24}$. There are four total measurement (measurements at four times of the day) and the problem is therefore overdetermined. Thus we determine the best fit amplitude and phase using a "sliding profile approach", as depicted in Fig 2 and described below."
**AC:** Strictly mathematically, the problem of two variables and four measurement points is overdetermined. However, with the shape as in Fig. 2a and realistic measurement errors one can easily show that 2 measurements per day are not enough to retrieve the parameters of the diurnal cycle (e.g. one can fix the first and the third points at 0.5 and the positive and negative phases would give the same solution). The suggested phrase about the "overdetermined problem" can create a feeling that the information coming from the second satellite is redundant.
To avoid this, we modify the suggested text to "With the $A_{12}/A_{24}$ set to 0.28 and $\Delta\phi$ set to 0, the diurnal "shape" of Eq. (1) is fixed (see the grey line in Fig. 2a) and the

problem reduces to one of determining the amplitude $A_{24}$ and phase $\phi_{24}$. Two satellite instruments provide us with measurements four times of the day, and we determine the best fit amplitude and phase using a minimization technique based on the "sliding profile" approach as depicted in Fig. 2 and described below." We believe that this text informs the reader about all necessary details.

**RC:** As regards the description of Figure 2 and the fitting process:
Page 7, line 13-15. I don't follow this description. Please rephrase. Why should a cloud fraction value of 0.2 yield a zero over zero? Does this mean the first red point in Fig 2c (near 1 UTC) is not used in the amplitude calculation?
**AC:** We have rephrased the text. As for the cloud fraction of 0.2 – the A(t) which is discussed here comes from Eq. 1 where the function changes sign and the amplitude is determined with respect to some virtual "zero line". Correspondingly, taking the values close to this line and building their ratios will lead to errors and therefore should be avoided.

**RC:** It is in no sense obvious to me that your sliding profile approach is any better than doing a least squares fit or any other minimization approach. Not that I think that it needs to be better, per se, but if you have a specific rationale for developing this approach rather than using established techniques for over determined problems it would be good to explain such
**AC:** This is a good point in the sense that the information content of the measured signal is always the same. However, the classical minimization approach applied here would operate in 2D space with the same weights for both variables giving less control over the retrieval. In our approach, we separate the variables and give the first priority to the phase because we believe that this is more important for understanding the mechanisms driving the cloud formation. In this case, the sensitivity of the problem to phase is directly visualized as the first fitting parameter and this makes it easier to explain as we can tell from our experience of presenting it to colleagues at different

conferences. However, indeed, one should be able to retrieve close values using a different approach. We added an explanation to the text before the paragraph justifying the choice of the function.

**RC:** 2) Uncertainty in diurnal amplitude and phase due to ASSUME $A_{12}/A_{24}$ ratio and $\Delta\phi$. You assess the uncertainty in your results due to the impact of a 20% random uncertainty in the cloud fraction, the result of which is a 20% uncertainty in the amplitude (which is not surprising) and 1.5 hours uncertainty in the phase. This is a GOOD result! But what about the assumed values for $A_{12}/A_{24}$ and $\Delta\phi$? I can easily imagine that even a small change in either of these assumed values might have a large impact. In particular, I expect these values might have spatially coherent regional variations that might impact your later analysis.

**AC:** This is an absolutely valid question, the answer to which is partially given at the end of the section 2.2 where we discuss the harmonic fit. For the sake of simplicity we did not include in the manuscript the figure resulting from this numerical experiment, which we provide below (the 1st figure). The numbers in this plot tell us that the real data can be better fitted with Eq. 1 than with a simple harmonic fit. To demonstrate the sensitivity of the retrieved amplitude and phase to the assumptions, we also performed a numerical experiment where we tried to fit the original function given by Eq. 1 using the simulated functions, for which we varied $A_{12}/A_{24}$ in the limits from 0 to 0.5 and $\Delta\phi$ from 0 to 24 hours.

As one can see from the second and third figures attached to the answer, varying $A_{12}/A_{24}$ ratio up to 0.35 does not induce more than a 0.5 hour shift in the peak time definition for any of the $\Delta\phi$ value. The most unfortunate combinations of $A_{12}/A_{24}$ and $\Delta\phi$ result in up to 2 hour shift, but there is no reason to believe that these combinations are realistic given that Eq. 1 provides a good quality fit. In the same way, the retrieved amplitude error rarely reaches 10%, so if we combine these values with the ones caused by the random uncertainty in the cloud fraction, this will not significantly change the uncertainty estimates given in the manuscript (20% uncertainty turns to 22% and

1.5 hour uncertainty turns to 1.6 hours). If the reviewer prefers to have this information in the methodological part, we can update these numbers and add an explanation. We can also add some or all of the panels of the attached figures to the Supplement.

**RC:** 3) Uncertainty due to sampling variability. I am very surprised to see that you do not discuss the sampling variability. Is the sampling variability small relative to the diurnal variations? For example, if you plotted the same result using any randomly chosen 5 or 8 years of data (rather than all 10 or 15 years) would all the results be so close together that I couldn't see any differences in any of the plots or values presented in Tables 1 and 2?
**AC:** We believe that one has to distinguish the methodological uncertainty, which is based on monthly data and which is discussed above, from the uncertainty caused by averaging the diurnal cycles over the years, which we do not discuss for the sake of simplicity. Of course, the averaged plots shown in Section 3.2 will slightly change if we remove some of the years, but their purpose is to give a general overview of what to expect from our data, which will be provided independently for each month.

**RC:** 4) What is new here? Do the results of the diurnal analysis differ from results of prior observational studies or extend our understanding of the diurnal cycle in some way? While reproducibility of results is an important aspect of science and having some "new result" is not strictly necessary, I think the article would be more interesting if it does. In general, more concrete discussion on how the results are similar (or not) to other studies would make the analysis more valuable
**AC:** Indeed, there are no groundbreaking findings in this work, but we highlight the following features, which have not been done before: (i) retrieving a consistent cloud dataset from two instruments using the same principle, but different wavelengths, instrumental functions, footprints, etc. – this is not trivial because any issue in the cloud retrieval methodology will wash out any subtle effect like diurnal variation; (ii) the diurnal cycle retrieval approach from 4 measurements per day in the infrared is
also new; (iii) the results confirm the observations made by other satellite instruments (we have added a discussion and modified some plots) and are free of different day vs night sensitivity issues (see the discussion of CATS and ISCCP); (iv) the results show the correlation and lags between the elements of the convective system. The suggested detailed comparison with other instruments is not straightforward because the sensitivity of different instruments to cloud thickness is not the same (see GEWEX report, Stubenrauch et al., 2012). On the other hand, we validate our results using the data from the active sounder (CALIOP lidar) that gives confidence for the clouds with the emissivity in the 0.1–0.95 range and we also compare the zonal averages with diurnal cycle retrieved from ISCCP observations. In addition, in the new Section 3.4 we relate the amplitudes of diurnal cycle to climate fluctuations.

**Minor comments**

**RC:** Introduction, first paragraph: I know this is well known material, but if you are going to start off a publication with such, you ought to support the material with appropriate references. Frankly, I think you might just drop this first paragraph and start the introduction with the second paragraph "Due to the importance of clouds . . . " (line 3, page 2).
**AC:** We have re-written the introduction taking into account the recommendation.

**RC:** Page 2, line 5. Starting phrase is awkward, I suggest change to read simply, "The diurnal variation modulates . . .".
**AC:** The phrase has been modified as suggested, thanks.

**RC:** Page 2, line 8. Perhaps change to "Many early global observational analyses ... ".
**AC:** This text has been re-written.

**RC:** Page 2. Line 10. Perhaps change "are significant" to "were found to be".
**AC:** We have re-phrased this sentence.

**RC:** Page 2. Line 20-23. I think these three bullet points are also intended to apply to cloud over land in the tropics – and don't for example apply to mid-latitude in the winter. Perhaps move the phrase "over land in the tropics" from the first bullet into line 18, or otherwise clarify.
**AC:** Thanks for noticing this inconsistency. The text has been modified.

**RC:** Page 4, line 1. What does "transformed" mean here?
**AC:** We have removed this word since it was related to technical processing, which does not explain much to the reader.

**RC:** Section 2.1. The spatial resolution of AIRS and IASI is coarse compared with most IR imagers. Perhaps it would be good to include some comments on how resolution impacts the cloud classes: high opaque ($\varepsilon$cld > 0.95), cirrus (0.95 > $\varepsilon$cld > 0.5), and thin cirrus (0.5 > $\varepsilon$cld > 0.1).
**AC:** These concerns have been addressed in the articles we refer to in the manuscript, namely, Stubenrauch et al., 2010, 2017. Indeed, if the pixel is only partially covered with the cloud, the detected outgoing radiance represents a weighted sum of clear sky and cloud radiances, so we had to adjust the detection thresholds using nearly collocated CALIPSO and AIRS observations. With these adjustments, we obtained 85–95% hit rate for AIRS vs CALIOP cloud detection over land and oceans in the tropics and midlatitudes, which we consider to be a good result taking into account the difference in the footprint sizes of two instruments. In Fig. S3 of the supplement to Stubenrauch et al., 2017 we show the accuracy of cloud height detection versus the emissivity.

**RC:** Page 4, line 1. Also what does the correction to SST entail? If it is truly

unimportant to the study results, then perhaps leave this line out. If it does matter, then you should probably explain a bit more.

**AC:** Again, this has been discussed in details in the article we are referring to (Stubenrauch et al., 2017), here we just remind the reader that we are not simply taking the surface temperature database with a known lack of diurnal variation. Even though it is more important for low level cloud retrievals and for the diurnal cycle of these clouds, which is not within the scope of current work, we do take care of these details. If the reviewer prefers, we can leave this line out, but in our opinion it makes sense to keep it.

**RC:** Page 4, line 5. Perhaps use "pixels" or "fields of view" rather than "scenes". To me "scene" implies a collection of pixels on some scale, for example a 60 km scene. As such a scene might have high-cloud even if only a fraction of the pixels contain clouds.

**AC:** We see the point, but we do not use the term "pixel" as it is linked with the imagers. The AIRS terminology names a bunch of measurements taken simultaneously a "golf ball", which consists of nine "spots", and we perform an individual retrieval for each of the spots, but to avoid the confusion we opted to generalize to "observations".

**RC:** Page 5, line 6. The word "admixture" seems a bit archaic to me and in general the idea of the diurnal cycle being a mixture of semi-diurnal cycles seems odd. Perhaps change to "... since variations in cloud amounts are known to include variations on both diurnal and semi-diurnal time scales."

**AC:** We agree with the suggestion, thanks.

**RC:** Page 7, line 1. What do you mean by "hit rate"? How does this translate into a 20% random noise?

**AC:** "Hit rate of X %" means that in X % of the collocated cases the cloud detected (or not detected) by AIRS was detected (or not detected) by CALIOP (hit rate is determined as a ratio of $(AIRS_{CLOUD};CALIPSO_{CLOUD})+(AIRS_{CLEAR};CALIPSO_{CLEAR})/total_N$).

Correspondingly, if we take a pessimistic estimate of the probability of a wrong detection (100% minus the numbers of Stubenrauch et al., 2017), it should not exceed 20%. We added a brief explanation to the end of the paragraph.

**RC:** Page 7, line 3. Perhaps change "build the function" to "solve for $A_{24}$ and $\phi_{24}$". Also as far as I can see, you are only looking at high clouds in this study, so $\Delta\phi$ is simply 0.
**AC:** We have added the comment regarding the $\Delta\phi$=0, but the suggested modification at the beginning the sentence does not match the algorithm: we build the function and then the solution is found in two steps. First, we move the function looking for the best phase shift and then we estimate the amplitude.

**RC:** Page 7, line 4. Perhaps change to "Having solved for $\phi_{24}$, ..."
**AC:** We modified the sentence to "Then we numerically solve the system for $\phi_{24}$" to continue the algorithm and to specify that we are solving for two parameters.

**RC:** Page 7, line 13-15. I don't follow this description. Please rephrase. Why should a cloud fraction value of 0.2 yield a zero over zero?
**AC:** Please, see the answer to the same question in General comments. We have updated the text to make it clearer.

**RC:** Page 8, line 3. Change "... separately of high opaque ..." to "... for high opaque ..."
**AC:** We have changed the text.

**RC:** Figure 3. What is being "averaged" in Figure 3? Are the data being spatially averaged (if so what was the starting spatial scale)? Or temporally averaged (what time period)? For example do you first calculate the diurnal amplitude and peak time for each 1x1 latitude region (for each cloud type) using all years of data and then

average spatially, or using 1 year of data and average in space and time ?

**AC:** For each zone, we first take the phase and amplitude value for a 1 lat x 1 lon monthly box already found and stored in our database, build the functions in accordance with Eq. 1 over 24 h with fine step, and average them. The resulting curve, therefore, represents both temporal and spatial average. We have added a brief description of the averaging procedure to the text.

**RC:** Page 8, line 14. Early studies. Please provide specific references.

**AC:** We have added several references.

**RC:** Page 8, line 20. I presume you mean with the estimate 1.5 hours uncertainty. Perhaps clarify. (As a very minor point I note that 0 h to 22 h is a 2 hour difference, so perhaps just say, they agree to within 2 hours).

**AC:** Both statements are correct, so we have modified the text accordingly.

**RC:** Top of Page 9. What does "In situ Freezing" mean?

**AC:** By using this term we tried to distinguish the ice formation due to cooling of the upwelling humid air from the freezing of already existing water vapor due to local temperature decrease. To avoid the confusion, we changed the text to "some of these thin cirrus are formed locally having no relation to the convective systems."

**RC:** Page 8 / figure 4 Discussion. A) What about mid-latitudes? Don't they deserve some discussion? In particular, (A1) Over land, why are cirrus and thin cirrus cloud in phase and LEAD opaque cloud? This is very different than tropical/subtropical land. (A2) Over midlatitude land, diurnal variation of thin cirrus is larger than cirrus while the opposite is true for subtropics and tropics (where cirrus larger than thin cirrus). This is worth a comment, I think. (C3) Over mid-latitude ocean, are the relatively small variations real (larger than uncertainty)? If yes, does this mean that cirrus form more prominently overnight and thin during the day or might this be a

retrieval artifact or ??

**AC:** This is a good point, but we don't think that one should compare the phases of the contribution curves and make the conclusions using only them. We provide these curves mostly to show the amplitude components of the total high cloud amount which we compare with the values retrieved from CATS and ISCCP whereas averaging the phase is a more subtle thing. Below in the text of the manuscript, we show that even for smaller areas the phase/amplitude diagrams (Fig. 9) are sometimes washed out and the phases of individual $1° \times 1°$ spots vary within several hours. When it comes to phases, we put more confidence in the individual $1° \times 1°$ values we show in Fig. 5-7 and in zonal averages calculated using a "resulting force" algorithm. In the present version of the manuscript we also have an updated discussion of Fig. 3 and Fig. 4.

**RC:** B) Subtropics ocean is similar to tropical ocean and subtropical land is similar to tropical land for boreal summer. This doesn't seem too surprising. What about boreal winter? Are subtropics similar to tropics in boreal winter? Similar to midlatitudes? Or differ from both? (Perhaps put NH winter and SH winter plots in supplementary material).

**AC:** Sorry, we didn't get this question – Fig. S1 of the supplement already shows the zones in the winter hemisphere and we mention this in the original text of the manuscript saying that the "amplitudes are close to zero in winter hemisphere". As one can see both from these plots and from Fig. 5 and 6, there's almost no material for the discussion for these zones.

**RC:** C) In general, is there anything here that is new or different from previous studies?

**AC:** The idea of building the CATS-like plots was twofold: (a) to validate our joint cloud retrieval + diurnal cycle retrieval methodology using the data from the active sounder and (b) to take advantage of the analysis performed by (Noel et al., 2018), who compared the diurnal cycle retrieved from the CATS observations with geostationary

and ground-based observations. Basically, we show the consistency of our data with those of CATS and assume that we can extend the conclusions of (Noel et al., 2018) to our work. We also discuss and explain the similarities and differences between our results, CATS, and ISCCP diurnal cycles in this section.

**RC:** Page 11, line 2. 0.85 seems an arbitrary choice. I presume your results are NOT sensitive to this choice, meaning if you choose 0.7 it has little impact ? Please comment.

**AC:** The results cannot be insensitive to the choice because it is related to the goodness-of-fit and fitting 4 points with Eq. 1 and realistic noise resulting in $k_{corr}$ equal to, let's say, 0.3 will never be equal to the fitting with 0.9. In the latter case, we have much more confidence that the retrieved diurnal cycle is not just a random function barely fitting the noise. Of course, $k_{corr}$=0.3 is an exaggeration and indeed the threshold of 0.7 still filters out pure noise, but the figures like, e.g. Fig. 5 become more cluttered. So, we have chosen a threshold, $k_{corr}$ which guarantees the presence of all main features and at the same time keeps the image "tidy". We understand that this is a subjective criterion, that's why we provide A$_{24}$, $\phi_{24}$, and $k_{corr}$ in the database, so that the researcher could make his/her own filtering, if needed.

**RC:** Page 11, line 10. Are the differences with Hong ... Random? Systematic is some regions? Earlier or later?

**AC:** We have updated the comparison of AIRS/IASI diurnal cycle with Hong et al., 2006.

**RC:** Page 11, line 22. On stability. I agree that there are many fewer points with a retrieved diurnal cycle in the winter hemisphere but there are some signals that perhaps deserve some comments. In Figure 5a, is the feature in the (A) Atlantic off the US East Coast, (B) the Central Pacific, and (C) that Himalayas real? (Do other climatologies show these features? Any idea what is going on here?)). Likewise, I

wonder if the diurnal cycle observed of the Southern Ocean is real.

**AC:** Since we used a uniform filtering on $k_{corr}$ (see the answer to the "Page 11, line 2" question) and the areas under consideration are quite large we should consider the detected diurnal variation in these regions as real as in the other ones. Similar features have been reported by e.g. Soden et al., 2000 or by Eastman and Warren, 2014, but taking into account smaller diurnal cycle amplitudes over the ocean, we would abstain from the discussion of these features on a $1°{\times}1°$ scale. Perhaps, a finer tuning of the detection threshold is needed for their analysis, but we wanted to use a uniform set of rules for the whole globe. As for Himalayas, the features are similar to those reported in Eastman and Warren, 2014 for the convective clouds (see their Fig. 8). We have added a short discussion on the detection of diurnal cycle over the oceans at the end of this paragraph.

**RC:** Page 11, line 26. Perhaps change to "In July, there is a large contrast between continental opaque clouds and nearby oceanic regions. Over the continents, the peak typically often occurs in the evening around 20h as compared with oceanic areas near the continents with peaks closer to noon. Nonetheless, in some other ocean locations opaque clouds also peak in the evening or overnight (e.g. tropical longitudes -115 to -135)."

**AC:** We added the suggested fragment, thanks.

**RC:** Page 11, line 27. As regards Indonesia, I think the Maritime continent has a complex land/sea breeze interactions that I don't think the 1 degree data can capture well. I suggest it might be better to simply note this (and perhaps cite appropriate references) rather than trying to make any conclusions for this region.

**AC:** We have rewritten this part of the text.

**RC:** Figure 8. I suggest changing the numbering scheme used in Figure 8 and the discussion to match the order of the regions shown in Figure 9 and tables 1 and 2

from top to bottom. This will make the text easier to follow.

**AC:** We followed the suggestion and renumbered the regions in Fig. 8 and in the text.

**RC:** Section 3.3. What guided the selection of regions analyzed here? In particular, why are no oceanic regions selected for analysis? Also, the North America region that is highlighted includes the west coast and Rockies but misses most of the central U.S. and Eastern U.S. Was the intent to avoid Mesoscale Convective Systems (MCSs) that strongly influence the diurnal phase and amplitude in the Central/Southern US? I note that Figure 6a shows the Central and Eastern US is notably different from the western US.

**AC:** The selection was a tradeoff between the areas of observed strong variability with a stable phase in winter and summer for all (or at least for few) cloud types. The idea was to use the same regions to compare winter and summer diurnal cycles thus reducing the number of elements in the tables. As for the oceans, there was no region which would satisfy the aforementioned criteria and the phase/amplitude diagrams built for ocean areas did not allow to make a sound conclusion, so we decided to exclude these regions. Averaging over a large ocean zone (Fig. 3) improves the statistics and allows tracking some features, but the size of the region is considerably larger than any of the land regions. As for the Western/Central US, the choice was primarily driven by the features seen in Fig. 5-7b,c and B1. Increasing the size of the region would lead to mixing up several areas with different phases and make the phase/amplitude diagrams unusable.

**RC:** Page 15, line 8. I agree that multiple peaks could be explained by orographic effects, however, I also think that MCSs likely play some role here. In general, what leads you to make this statement? Is the intent that this statement is a conclusion Or are you speculating?

**AC:** It's true, there is no bullet-proof evidence of the mechanism responsible for these effects, and this is just one of possible explanations, so we toned this statement down

to: "multiple peak amplitude local times may indicate effects of orography".

**RC:** Figure 9. Suggest change column header "Conv." To "Opq" in keeping with the figure caption, and to be consistent with the point that Opaque is not perfectly synonymous with convection.
**AC:** Thanks for noticing this. This has been fixed.

**RC:** Page 17, line 10. As with earlier comment, I do not know what "in situ freezing" means.
**AC:** Please, see the comment to "Top of Page 9" question. We have updated the text on p.17.

**RC:** Page 18, line 3. I am not clear on what "in the same limits" means here. Please rephrase.
**AC:** We have rephrased it to "Even though peak times for cirrus clouds and for high opaque clouds vary almost in the same range, an average lag of 3 h can be identified for two thirds of the cases".

**RC:** Page 19, line 4-6. You write, "It is interesting to note that the local time of the minimum moves from the summer hemisphere midlatitudes towards the tropics from 6h to noon and the one of the maximum from 17h to 1h." I am confused, what is moving, the total amount of upper level cloud? What does "one of the maximum" mean? Perhaps expand and/or rephrase this remark.
**AC:** It is the local time of the minimum and maximum cloud amount that moves, but we decided to remove this sentence because it confuses the reader and contradicts with the methodology used in the study, according to which these times can be relatively well determined even if the local times of the observations do not match the maximum or the minimum. We rephrased this sentence to "AIRS alone with its observations at 1h30 and 13h30 LT is closer to capturing the maximum and minimum of cloud cover

over tropical land than IASI observing the atmosphere at 9h30 and 21h30 LT."

**RC:** Page 19, line 10. What other analyses? Suggest references. Is there anything in the present work which adds to OR departs from the "other analysis?
**AC:** Please, see the answer to the general question 4, to the question to page 11, line 10, and the discussion around Fig. 3 and 4. Since we have added more references and updated the discussion above (e.g. see the corresponding references in Section 3.2.) we do not include them here to keep the flow of this section uninterrupted.

[Figure]

**Fig. 1.** Justifying the choice of the fitting function

peak time difference w.r.t. original [hours]

Fig. 2. Estimating the effects of A12/A24 and delta phi uncertainties on peak time definition

Amplitude ratio w.r.t. original

**Fig. 3.** Estimating the effects of A12/A24 and delta phi uncertainties on the retrieved amplitude

---

## Author Response (AR2)

We thank the Reviewer # 1 for his/her comments on the revised version of the manuscript.

We have fixed the minor issue detected by the reviewer and we added the "Data availability" and "Competing interests" sections to the manuscript.